# Fast Algorithms for Robust PCA via Gradient Descent

**Xinyang Yi**[*]    **Dohyung Park**[*]    **Yudong Chen**[†]    **Constantine Caramanis**[*]
[*]The University of Texas at Austin    [†]Cornell University
[*]{yixy,dhpark,constantine}@utexas.edu    [†]yudong.chen@cornell.edu

## Abstract

We consider the problem of Robust PCA in the fully and partially observed settings. Without corruptions, this is the well-known matrix completion problem. From a statistical standpoint this problem has been recently well-studied, and conditions on when recovery is possible (how many observations do we need, how many corruptions can we tolerate) via polynomial-time algorithms is by now understood. This paper presents and analyzes a non-convex optimization approach that greatly reduces the computational complexity of the above problems, compared to the best available algorithms. In particular, in the fully observed case, with $r$ denoting rank and $d$ dimension, we reduce the complexity from $\mathcal{O}(r^2 d^2 \log(1/\varepsilon))$ to $\mathcal{O}(r d^2 \log(1/\varepsilon))$ – a big savings when the rank is big. For the partially observed case, we show the complexity of our algorithm is no more than $\mathcal{O}(r^4 d \log d \log(1/\varepsilon))$. Not only is this the best-known run-time for a provable algorithm under partial observation, but in the setting where $r$ is small compared to $d$, it also allows for near-linear-in-$d$ run-time that can be exploited in the fully-observed case as well, by simply running our algorithm on a subset of the observations.

## 1  Introduction

*Principal component analysis* (PCA) aims to find a low rank subspace that best-approximates a data matrix $Y \in \mathbb{R}^{d_1 \times d_2}$. The simple and standard method of PCA by *singular value decomposition* (SVD) fails in many modern data problems due to missing and corrupted entries, as well as sheer scale of the problem. Indeed, SVD is highly sensitive to outliers by virtue of the squared-error criterion it minimizes. Moreover, its running time scales as $\mathcal{O}(r d^2)$ to recover a rank $r$ approximation of a $d$-by-$d$ matrix.

While there have been recent results developing provably robust algorithms for PCA (e.g., [5, 26]), the running times range from $\mathcal{O}(r^2 d^2)$ to $\mathcal{O}(d^3)$ and hence are significantly worse than SVD. Meanwhile, the literature developing sub-quadratic algorithms for PCA (e.g., [15, 14, 3]) seems unable to guarantee robustness to outliers or missing data.

Our contribution lies precisely in this area: provably robust algorithms for PCA with improved run-time. Specifically, we provide an efficient algorithm with running time that matches SVD while nearly matching the best-known robustness guarantees. In the case where rank is small compared to dimension, we develop an algorithm with running time that is nearly linear in the dimension. This last algorithm works by subsampling the data, and therefore we also show that our algorithm solves the Robust PCA problem with partial observations (a generalization of matrix completion and Robust PCA).

### 1.1  The Model and Related Work

We consider the following setting for robust PCA. Suppose we are given a matrix $Y \in \mathbb{R}^{d_1 \times d_2}$ that has decomposition $Y = M^* + S^*$, where $M^*$ is a rank $r$ matrix and $S^*$ is a sparse corruption matrix containing entries with arbitrary magnitude. The goal is to recover $M^*$ and $S^*$ from $Y$. To ease notation, we let $d_1 = d_2 = d$ in the remainder of this section.

Provable solutions for this model are first provided in the works of [9] and [5]. They propose to solve this problem by *convex relaxation*:

$$\min_{M,S} \|M\|_{\text{nuc}} + \lambda\|S\|_1, \text{ s.t. } Y = M + S, \tag{1}$$

where $\|M\|_{\text{nuc}}$ denotes the nuclear norm of $M$. Despite analyzing the same method, the corruption models in [5] and [9] differ. In [5], the authors consider the setting where the entries of $M^*$ are corrupted at random with probability $\alpha$. They show their method succeeds in exact recovery with $\alpha$ as large as $0.1$, which indicates they can tolerate a constant fraction of corruptions. Work in [9] considers a *deterministic corruption model*, where nonzero entries of $S^*$ can have arbitrary position, but the sparsity of each row and column does not exceed $\alpha d$. They prove that for exact recovery, it can allow $\alpha = \mathcal{O}(1/(\mu r\sqrt{d}))$. This was subsequently further improved to $\alpha = \mathcal{O}(1/(\mu r))$, which is in fact optimal [11, 18]. Here, $\mu$ represents the incoherence of $M^*$ (see Section 2 for details). In this paper, we follow this latter line and focus on the deterministic corruption model.

The state-of-the-art solver [20] for (1) has time complexity $\mathcal{O}(d^3/\varepsilon)$ to achieve error $\varepsilon$, and is thus much slower than SVD, and prohibitive for even modest values of $d$. Work in [21] considers the deterministic corruption model, and improves this running time without sacrificing the robustness guarantee on $\alpha$. They propose an *alternating projection* (AltProj) method to estimate the low rank and sparse structures iteratively and simultaneously, and show their algorithm has complexity $\mathcal{O}(r^2d^2\log(1/\varepsilon))$, which is faster than the convex approach but still slower than SVD.

Non-convex approaches have recently seen numerous developments for applications in low-rank estimation, including alternating minimization (see e.g. [19, 17, 16]) and gradient descent (see e.g. [4, 12, 23, 24, 29, 30]). These works have fast running times, yet do not provide robustness guarantees. One exception is [12], where the authors analyze a row-wise $\ell_1$ projection method for recovering $S^*$. Their analysis hinges on positive semidefinite $M^*$, and the algorithm requires prior knowledge of the $\ell_1$ norm of every row of $S^*$ and is thus prohibitive in practice. Another exception is work [16], which analyzes alternating minimization plus an overall sparse projection. Their algorithm is shown to tolerate at most a fraction of $\alpha = \mathcal{O}(1/(\mu^{2/3}r^{2/3}d))$ corruptions. As we discuss in Section 1.2, we can allow $S^*$ to have much higher sparsity $\alpha = \mathcal{O}(1/(\mu r^{1.5}))$, which is close to optimal. It is worth mentioning other works that obtain provable guarantees of non-convex algorithms or problems including phase retrieval [6, 13, 28], EM algorithms [2, 25, 27], tensor decompositions [1] and second order method [22]. It might be interesting to bring robust considerations to these works.

### 1.2 Our Contributions

In this paper, we develop efficient non-convex algorithms for robust PCA. We propose a novel algorithm based on the projected gradient method on the factorized space. We also extend it to solve robust PCA in the setting with partial observations, i.e., in addition to gross corruptions, the data matrix has a large number of missing values. Our main contributions are summarized as follows.[1]

1. We propose a novel sparse estimator for the setting of deterministic corruptions. For the low-rank structure to be identifiable, it is natural to assume that deterministic corruptions are "spread out" (no more than some number in each row/column). We leverage this information in a simple but critical algorithmic idea, that is tied to the ultimate complexity advantages our algorithm delivers.

2. Based on the proposed sparse estimator, we propose a projected gradient method on the matrix factorized space. While non-convex, the algorithm is shown to enjoy linear convergence under proper initialization. Along with a new initialization method, we show that robust PCA can be solved within complexity $\mathcal{O}(rd^2\log(1/\varepsilon))$ while ensuring robustness $\alpha = \mathcal{O}(1/(\mu r^{1.5}))$. Our algorithm is thus faster than the best previous known algorithm by a factor of $r$, and enjoys superior empirical performance as well.

3. Algorithms for Robust PCA with partial observations still rely on a computationally expensive convex approach, as apparently this problem has evaded treatment by non-convex methods. We consider precisely this problem. In a nutshell, we show that our gradient method succeeds (it is guaranteed to produce the subspace of $M^*$) even when run on no more than $\mathcal{O}(\mu^2r^2d\log d)$ random entries of $Y$. The computational cost is $\mathcal{O}(\mu^3r^4d\log d\log(1/\varepsilon))$. When rank $r$ is small compared to the dimension $d$, in fact this dramatically improves on our bound above, as our cost becomes nearly linear in $d$. We show, moreover, that this savings and robustness to erasures comes at *no cost in the*

*robustness guarantee* for the deterministic (gross) corruptions. While this demonstrates our algorithm is robust to both outliers and erasures, it also provides a way to reduce computational costs even in the fully observed setting, when $r$ is small.

4. An immediate corollary of the above result provides a guarantee for exact matrix completion, with general rectangular matrices, using $\mathcal{O}(\mu^2 r^2 d \log d)$ observed entries and $\mathcal{O}(\mu^3 r^4 d \log d \log(1/\varepsilon))$ time, thereby improving on existing results in [12, 23].

**Notation.** For any index set $\Omega \subseteq [d_1] \times [d_2]$, we let $\Omega_{(i,\cdot)} := \{(i,j) \in \Omega \mid j \in [d_2]\}$, $\Omega_{(\cdot,j)} := \{(i,j) \in \Omega \mid i \in [d_1]\}$. For any matrix $A \in \mathbb{R}^{d_1 \times d_2}$, we denote its projector onto support $\Omega$ by $\Pi_\Omega(A)$, i.e., the $(i,j)$-th entry of $\Pi_\Omega(A)$ is equal to $A$ if $(i,j) \in \Omega$ and zero otherwise. The $i$-th row and $j$-th column of $A$ are denoted by $A_{(i,\cdot)}$ and $A_{(\cdot,j)}$. The $(i,j)$-th entry is denoted as $A_{(i,j)}$. Operator norm of $A$ is $\|A\|_{\mathrm{op}}$. Frobenius norm of $A$ is $\|A\|_{\mathrm{F}}$. The $\ell_a/\ell_b$ norm of $A$ is denoted by $\|A\|_{b,a}$, i.e., the $\ell_a$ norm of the vector formed by the $\ell_b$ norm of every row. For instance, $\|A\|_{2,\infty}$ stands for $\max_{i \in [d_1]} \|A_{(i,\cdot)}\|_2$.

## 2 Problem Setup

We consider the problem where we observe a matrix $Y \in \mathbb{R}^{d_1 \times d_2}$ that satisfies $Y = M^* + S^*$, where $M^*$ has rank $r$, and $S^*$ is corruption matrix with sparse support. Our goal is to recover $M^*$ and $S^*$. In the partially observed setting, in addition to sparse corruptions, we have erasures. We assume that each entry of $M^* + S^*$ is revealed independently with probability $p \in (0,1)$. In particular, for any $(i,j) \in [d_1] \times [d_2]$, we consider the Bernoulli model where

$$Y_{(i,j)} = \begin{cases} (M^* + S^*)_{(i,j)}, & \text{with probability } p; \\ *, & \text{otherwise.} \end{cases} \tag{2}$$

We denote the support of $Y$ by $\Phi = \{(i,j) \mid Y_{(i,j)} \neq *\}$. Note that we assume $S^*$ is not adaptive to $\Phi$. As is well-understood thanks to work in matrix completion, this task is impossible in general – we need to guarantee that $M^*$ is not both low-rank and sparse. To avoid such identifiability issues, we make the following standard assumptions on $M^*$ and $S^*$: (i) $M^*$ is not near-sparse or "spiky." We impose this by requiring $M^*$ to be $\mu$-incoherent – given a singular value decomposition (SVD) $M^* = L^* \Sigma^* R^{*\top}$, we assume that

$$\|L^*\|_{2,\infty} \leq \sqrt{\frac{\mu r}{d_1}}, \quad \|R^*\|_{2,\infty} \leq \sqrt{\frac{\mu r}{d_2}}.$$

(ii) The entries of $S^*$ are "spread out" – for $\alpha \in [0,1)$, we assume $S^* \in \mathcal{S}_\alpha$, where

$$\mathcal{S}_\alpha := \left\{ A \in \mathbb{R}^{d_1 \times d_2} \mid \|A_{(i,\cdot)}\|_0 \leq \alpha d_2 \text{ for all } i \in [d_1] \,; \|A_{(\cdot,j)}\|_0 \leq \alpha d_1 \text{ for all } j \in [d_2] \right\}. \tag{3}$$

In other words, $S^*$ contains at most $\alpha$-fraction nonzero entries per row and column.

## 3 Algorithms

For both the full and partial observation settings, our method proceeds in two phases. In the first phase, we use a new sorting-based sparse estimator to produce a rough estimate $S_{\mathrm{init}}$ for $S^*$ based on the observed matrix $Y$, and then find a rank $r$ matrix factorized as $U_0 V_0^\top$ that is a rough estimate of $M^*$ by performing SVD on $(Y - S_{\mathrm{init}})$. In the second phase, given $(U_0, V_0)$, we perform an iterative method to produce series $\{(U_t, V_t)\}_{t=0}^\infty$. In each step $t$, we first apply our sparse estimator to produce a sparse matrix $S_t$ based on $(U_t, V_t)$, and then perform a projected gradient descent step on the low-rank factorized space to produce $(U_{t+1}, V_{t+1})$. This flow is the same for full and partial observations, though a few details differ. Algorithm 1 gives the full observation algorithm, and Algorithm 2 gives the partial observation algorithm. We now describe the key details of each algorithm.

**Sparse Estimation.** A natural idea is to keep those entries of residual matrix $Y - M$ that have large magnitude. At the same time, we need to make use of the dispersed property of $\mathcal{S}_\alpha$ that every column and row contain at most $\alpha$-fraction of nonzero entries. Motivated by these two principles, we introduce the following sparsification operator: For any matrix $A \in \mathbb{R}^{d_1 \times d_2}$: for all $(i,j) \in [d_1] \times [d_2]$, we let

$$\mathcal{T}_\alpha[A] := \begin{cases} A_{(i,j)}, & \text{if } |A_{(i,j)}| \geq |A_{(i,\cdot)}^{(\alpha d_2)}| \text{ and } |A_{(i,j)}| \geq |A_{(\cdot,j)}^{(\alpha d_1)}| \,, \\ 0, & \text{otherwise} \end{cases} \tag{4}$$

where $A_{(i,\cdot)}^{(k)}$ and $A_{(\cdot,j)}^{(k)}$ denote the elements of $A_{(i,\cdot)}$ and $A_{(\cdot,j)}$ that have the $k$-th largest magnitude respectively. In other words, we choose to keep those elements that are simultaneously among the largest $\alpha$-fraction entries in the corresponding row and column. In the case of entries having identical magnitude, we break ties arbitrarily. It is thus guaranteed that $\mathcal{T}_\alpha [A] \in \mathcal{S}_\alpha$.

---

**Algorithm 1** Fast RPCA

---

**INPUT:** Observed matrix $Y$ with rank $r$ and corruption fraction $\alpha$; parameters $\gamma, \eta$; number of iterations $T$.

   *// Phase I: Initialization.*
1: $S_{\text{init}} \leftarrow \mathcal{T}_\alpha [Y]$      *// see (4) for the definition of $\mathcal{T}_\alpha [\cdot]$.*
2: $[L, \Sigma, R] \leftarrow \text{SVD}_r [Y - S_{\text{init}}]$ [2]
3: $U_0 \leftarrow L\Sigma^{1/2}$, $V_0 \leftarrow R\Sigma^{1/2}$. Let $\mathcal{U}, \mathcal{V}$ be defined according to (7).

   *// Phase II: Gradient based iterations.*
4: $U_0 \leftarrow \Pi_{\mathcal{U}} (U_0), V_0 \leftarrow \Pi_{\mathcal{V}} (V_0)$
5: **for** $t = 0, 1, \dots, T - 1$ **do**
6:     $S_t \leftarrow \mathcal{T}_{\gamma\alpha} \left[ Y - U_t V_t^\top \right]$
7:     $U_{t+1} \leftarrow \Pi_{\mathcal{U}} \left( U_t - \eta \nabla_U \mathcal{L}(U_t, V_t; S_t) - \frac{1}{2}\eta U_t(U_t^\top U_t - V_t^\top V_t) \right)$
8:     $V_{t+1} \leftarrow \Pi_{\mathcal{V}} \left( V_t - \eta \nabla_V \mathcal{L}(U_t, V_t; S_t) - \frac{1}{2}\eta V_t(V_t^\top V_t - U_t^\top U_t) \right)$
9: **end for**
**OUTPUT:** $(U_T, V_T)$

---

**Initialization.** In the *fully observed* setting, we compute $S_{\text{init}}$ based on $Y$ as $S_{\text{init}} = \mathcal{T}_\alpha [Y]$. In the *partially observed* setting with sampling rate $p$, we let $S_{\text{init}} = \mathcal{T}_{2p\alpha} [Y]$. In both cases, we then set $U_0 = L\Sigma^{1/2}$ and $V_0 = R\Sigma^{1/2}$, where $L\Sigma R^\top$ is an SVD of the best rank $r$ approximation of $Y - S_{\text{init}}$.

**Gradient Method on Factorized Space.** After initialization, we proceed by projected gradient descent. To do this, we define loss functions explicitly in the factored space, i.e., in terms of $U, V$ and $S$:

$$\mathcal{L}(U, V; S) := \frac{1}{2}\|UV^\top + S - Y\|_{\text{F}}^2, \qquad \text{(fully observed)} \qquad (5)$$

$$\widetilde{\mathcal{L}}(U, V; S) := \frac{1}{2p}\|\Pi_\Phi \left( UV^\top + S - Y \right)\|_{\text{F}}^2. \qquad \text{(partially observed)} \qquad (6)$$

Recall that our goal is to recover $M^*$ that satisfies the $\mu$-incoherent condition. Given an SVD $M^* = L^*\Sigma R^{*\top}$, we expect that the solution $(U, V)$ is close to $(L^*\Sigma^{1/2}, R^*\Sigma^{1/2})$ up to some rotation. In order to serve such $\mu$-incoherent structure, it is natural to put constraints on the row norms of $U, V$ based on $\|M^*\|_{\text{op}}$. As $\|M^*\|_{\text{op}}$ is unavailable, given $U_0, V_0$ computed in the first phase, we rely on the sets $\mathcal{U}, \mathcal{V}$ defined as

$$\mathcal{U} := \left\{ A \in \mathbb{R}^{d_1 \times r} \mid \|A\|_{2,\infty} \leq \sqrt{\frac{2\mu r}{d_1}} \|U_0\|_{\text{op}} \right\}, \ \mathcal{V} := \left\{ A \in \mathbb{R}^{d_2 \times r} \mid \|A\|_{2,\infty} \leq \sqrt{\frac{2\mu r}{d_2}} \|V_0\|_{\text{op}} \right\}. \tag{7}$$

Now we consider the following optimization problems with constraints:

$$\min_{U \in \mathcal{U}, V \in \mathcal{V}, S \in \mathcal{S}_\alpha} \mathcal{L}(U, V; S) + \frac{1}{8}\|U^\top U - V^\top V\|_{\text{F}}^2, \qquad \text{(fully observed)} \qquad (8)$$

$$\min_{U \in \mathcal{U}, V \in \mathcal{V}, S \in \mathcal{S}_{p\alpha}} \widetilde{\mathcal{L}}(U, V; S) + \frac{1}{64}\|U^\top U - V^\top V\|_{\text{F}}^2. \qquad \text{(partially observed)} \qquad (9)$$

The regularization term in the objectives above is used to encourage that $U$ and $V$ have the same scale. Given $(U_0, V_0)$, we propose the following iterative method to produce series $\{(U_t, V_t)\}_{t=0}^\infty$ and $\{S_t\}_{t=0}^\infty$. We give the details for the fully observed case – the partially observed case is similar.

For $t = 0, 1, \ldots$, we update $S_t$ using the sparse estimator $S_t = \mathcal{T}_{\gamma\alpha}\left[Y - U_t V_t^\top\right]$, followed by a projected gradient update on $U_t$ and $V_t$:

$$U_{t+1} = \Pi_{\mathcal{U}}\left(U_t - \eta\nabla_U \mathcal{L}(U_t, V_t; S_t) - \frac{1}{2}\eta U_t(U_t^\top U_t - V_t^\top V_t)\right),$$

$$V_{t+1} = \Pi_{\mathcal{V}}\left(V_t - \eta\nabla_V \mathcal{L}(U_t, V_t; S_t) - \frac{1}{2}\eta V_t(V_t^\top V_t - U_t^\top U_t)\right).$$

Here $\alpha$ is the model parameter that characterizes the corruption fraction, $\gamma$ and $\eta$ are algorithmic tunning parameters, which we specify in our analysis. Essentially, the above algorithm corresponds to applying projected gradient method to optimize (8), where $S$ is replaced by the aforementioned sparse estimator in each step.

---

**Algorithm 2** Fast RPCA with partial observations

**INPUT:** Observed matrix $Y$ with support $\Phi$; parameters $\tau, \gamma, \eta$; number of iterations $T$.

  // *Phase I: Initialization.*
  1: $S_{\text{init}} \leftarrow \mathcal{T}_{2p\alpha}\left[\Pi_\Phi(Y)\right]$
  2: $[L, \Sigma, R] \leftarrow \text{SVD}_r[\frac{1}{p}(Y - S_{\text{init}})]$
  3: $U_0 \leftarrow L\Sigma^{1/2}, V_0 \leftarrow R\Sigma^{1/2}$. Let $\mathcal{U}, \mathcal{V}$ be defined according to (7).
  // *Phase II: Gradient based iterations.*
  4: $U_0 \leftarrow \Pi_{\mathcal{U}}(U_0), V_0 \leftarrow \Pi_{\mathcal{V}}(V_0)$
  5: **for** $t = 0, 1, \ldots, T - 1$ **do**
  6: $\quad S_t \leftarrow \mathcal{T}_{\gamma p\alpha}\left[\Pi_\Phi\left(Y - U_t V_t^\top\right)\right]$
  7: $\quad U_{t+1} \leftarrow \Pi_{\mathcal{U}}\left(U_t - \eta\nabla_U \widetilde{\mathcal{L}}(U_t, V_t; S_t) - \frac{1}{16}\eta U_t(U_t^\top U_t - V_t^\top V_t)\right)$
  8: $\quad V_{t+1} \leftarrow \Pi_{\mathcal{V}}\left(V_t - \eta\nabla_V \widetilde{\mathcal{L}}(U_t, V_t; S_t) - \frac{1}{16}\eta V_t(V_t^\top V_t - U_t^\top U_t)\right)$
  9: **end for**
**OUTPUT:** $(U_T, V_T)$

---

## 4 Main Results

### 4.1 Analysis of Algorithm 1

We begin with some definitions and notation. It is important to define a proper error metric because the optimal solution corresponds to a manifold and there are many distinguished pairs $(U, V)$ that minimize (8). Given the SVD of the true low-rank matrix $M^* = L^*\Sigma^* R^{*\top}$, we let $U^* := L^*\Sigma^{*1/2}$ and $V^* := R^*\Sigma^{*1/2}$. We also let $\sigma_1^* \geq \sigma_2^* \geq \ldots \geq \sigma_r^*$ be sorted nonzero singular values of $M^*$, and denote the condition number of $M^*$ by $\kappa$, i.e., $\kappa := \sigma_1^*/\sigma_r^*$. We define estimation error $d(U, V; U^*, V^*)$ as the minimal Frobenius norm between $(U, V)$ and $(U^*, V^*)$ with respect to the optimal rotation, namely

$$d(U, V; U^*, V^*) := \min_{Q \in \mathbb{Q}_r} \sqrt{\|U - U^* Q\|_{\text{F}}^2 + \|V - V^* Q\|_{\text{F}}^2}, \tag{10}$$

for $\mathbb{Q}_r$ the set of $r$-by-$r$ orthonormal matrices. This metric controls reconstruction error, as

$$\|UV^\top - M^*\|_{\text{F}} \lesssim \sqrt{\sigma_1^*}\, d(U, V; U^*, V^*), \tag{11}$$

when $d(U, V; U^*, V^*) \leq \sqrt{\sigma_1^*}$. We denote the local region around the optimum $(U^*, V^*)$ with radius $\omega$ as

$$\mathbb{B}_2(\omega) := \left\{(U, V) \in \mathbb{R}^{d_1 \times r} \times \mathbb{R}^{d_2 \times r} \mid d(U, V; U^*, V^*) \leq \omega\right\}.$$

The next two theorems provide guarantees for the initialization phase and gradient iterations, respectively, of Algorithm 1.

**Theorem 1** (Initialization)**.** *Consider the paired* $(U_0, V_0)$ *produced in the first phase of Algorithm 1. If* $\alpha \leq 1/(16\kappa\mu r)$, *we have*

$$d(U_0, V_0; U^*, V^*) \leq 28\sqrt{\kappa}\alpha\mu r\sqrt{r}\sqrt{\sigma_1^*}.$$

**Theorem 2** (Convergence)**.** *Consider the second phase of Algorithm 1. Suppose we choose $\gamma = 2$ and $\eta = c/\sigma_1^*$ for any $c \leq 1/36$. There exist constants $c_1, c_2$ such that when $\alpha \leq c_1/(\kappa^2 \mu r)$, given any $(U_0, V_0) \in \mathbb{B}_2\left(c_2\sqrt{\sigma_r^*/\kappa}\right)$, the iterates $\{(U_t, V_t)\}_{t=0}^{\infty}$ satisfy*

$$d^2(U_t, V_t; U^*, V^*) \leq \left(1 - \frac{c}{8\kappa}\right)^t d^2(U_0, V_0; U^*, V^*).$$

Therefore, using proper initialization and step size, the gradient iteration converges at a linear rate with a constant contraction factor $1 - \mathcal{O}(1/\kappa)$. To obtain relative precision $\varepsilon$ compared to the initial error, it suffices to perform $O(\kappa \log(1/\varepsilon))$ iterations. Note that the step size is chosen according to $1/\sigma_1^*$. When $\alpha \lesssim 1/(\mu\sqrt{\kappa r^3})$, Theorem 1 and the inequality (11) together imply that $\|U_0 V_0^\top - M^*\|_{\text{op}} \leq \frac{1}{2}\sigma_1^*$. Hence we can set the step size as $\eta = \mathcal{O}(1/\sigma_1(U_0 V_0^\top))$ using being the top singular value $\sigma_1(U_0 V_0^\top)$ of the matrix $U_0 V_0^\top$

Combining Theorems 1 and 2 implies the following result that provides an overall guarantee for Algorithm 1.

**Corollary 1.** *Suppose that*

$$\alpha \leq c \min\left\{\frac{1}{\mu\sqrt{\kappa r^3}}, \frac{1}{\mu\kappa^2 r}\right\}$$

*for some constant c. Then for any $\varepsilon \in (0, 1)$, Algorithm 1 with $T = \mathcal{O}(\kappa \log(1/\varepsilon))$ outputs a pair $(U_T, V_T)$ that satisfies*

$$\|U_T V_T^\top - M^*\|_F \leq \varepsilon \cdot \sigma_r^*. \tag{12}$$

*Remark* 1 (Time Complexity)*.* For simplicity we assume $d_1 = d_2 = d$. Our sparse estimator (4) can be implemented by finding the top $\alpha d$ elements of each row and column via partial quick sort, which has running time $\mathcal{O}(d^2 \log(\alpha d))$. Performing rank-$r$ SVD in the first phase and computing the gradient in each iteration both have complexity $\mathcal{O}(rd^2)$.[3] Algorithm 1 thus has total running time $\mathcal{O}(\kappa r d^2 \log(1/\varepsilon))$ for achieving an $\epsilon$ accuracy as in (12). We note that when $\kappa = \mathcal{O}(1)$, our algorithm is orderwise faster than the AltProj algorithm in [21], which has running time $\mathcal{O}(r^2 d^2 \log(1/\varepsilon))$. Moreover, our algorithm only requires computing one singular value decomposition.

*Remark* 2 (Robustness)*.* Assuming $\kappa = \mathcal{O}(1)$, our algorithm can tolerate corruption at a sparsity level up to $\alpha = \mathcal{O}(1/(\mu r\sqrt{r}))$. This is worse by a factor $\sqrt{r}$ compared to the optimal statistical guarantee $1/(\mu r)$ obtained in [11, 18, 21]. This looseness is a consequence of the condition for $(U_0, V_0)$ in Theorem 2. Nevertheless, when $\mu r = \mathcal{O}(1)$, our algorithm can tolerate a constant $\alpha$ fraction of corruptions.

## 4.2 Analysis of Algorithm 2

We now move to the guarantees of Algorithm 2. We show here that not only can we handle partial observations, but in fact subsampling the data in the fully observed case can significantly reduce the time complexity from the guarantees given in the previous section without sacrificing robustness. In particular, for smaller values of $r$, the complexity of Algorithm 2 has *near linear dependence on the dimension $d$*, instead of quadratic.

In the following discussion, we let $d := \max\{d_1, d_2\}$. The next two results control the quality of the initialization step, and then the gradient iterations.

**Theorem 3** (Initialization, partial observations)**.** *Suppose the observed indices $\Phi$ follow the Bernoulli model given in (2). Consider the pair $(U_0, V_0)$ produced in the first phase of Algorithm 2. There exist constants $\{c_i\}_{i=1}^3$ such that for any $\epsilon \in (0, \sqrt{r}/(8c_1\kappa))$, if*

$$\alpha \leq \frac{1}{64\kappa\mu r}, \quad p \geq c_2 \left(\frac{\mu r^2}{\epsilon^2} + \frac{1}{\alpha}\right) \frac{\log d}{d_1 \wedge d_2}, \tag{13}$$

*then we have*

$$d(U_0, V_0; U^*, V^*) \leq 51\sqrt{\kappa}\alpha\mu r\sqrt{r}\sqrt{\sigma_1^*} + 7c_1\epsilon\sqrt{\kappa\sigma_1^*},$$

*with probability at least $1 - c_3 d^{-1}$.*

**Theorem 4** (Convergence, partial observations). *Suppose the observed indices $\Phi$ follow the Bernoulli model given in* (2). *Consider the second phase of Algorithm 2. Suppose we choose $\gamma = 3$, and $\eta = c/(\mu r \sigma_1^*)$ for a sufficiently small constant $c$. There exist constants $\{c_i\}_{i=1}^4$ such that if*

$$\alpha \leq \frac{c_1}{\kappa^2 \mu r} \quad \text{and} \quad p \geq c_2 \frac{\kappa^4 \mu^2 r^2 \log d}{d_1 \wedge d_2}, \tag{14}$$

*then with probability at least $1 - c_3 d^{-1}$, the iterates $\{(U_t, V_t)\}_{t=0}^\infty$ satisfy*

$$d^2(U_t, V_t; U^*, V^*) \leq \left(1 - \frac{c}{64 \mu r \kappa}\right)^t d^2(U_0, V_0; U^*, V^*)$$

*for all $(U_0, V_0) \in \mathbb{B}_2\left(c_4 \sqrt{\sigma_r^*/\kappa}\right)$.*

Setting $p = 1$ in the above result recovers Theorem 2 up to an additional factor $\mu r$ in the contraction factor. For achieving $\varepsilon$ relative accuracy, now we need $\mathcal{O}(\mu r \kappa \log(1/\varepsilon))$ iterations. Putting Theorems 3 and 4 together, we have the following overall guarantee for Algorithm 2.

**Corollary 2.** *Suppose that*

$$\alpha \leq c \min\left\{\frac{1}{\mu \sqrt{\kappa r^3}}, \frac{1}{\mu \kappa^2 r}\right\}, \quad p \geq c' \frac{\kappa^4 \mu^2 r^2 \log d}{d_1 \wedge d_2},$$

*for some constants $c, c'$. With probability at least $1 - \mathcal{O}(d^{-1})$, for any $\varepsilon \in (0, 1)$, Algorithm 2 with $T = \mathcal{O}(\mu r \kappa \log(1/\varepsilon))$ outputs a pair $(U_T, V_T)$ that satisfies*

$$\|U_T V_T^\top - M^*\|_F \leq \varepsilon \cdot \sigma_r^*. \tag{15}$$

This result shows that partial observations do not compromise robustness to sparse corruptions: as long as the observation probability $p$ satisfies the condition in Corollary 2, Algorithm 2 enjoys the same robustness guarantees as the method using all entries. Below we provide two remarks on the sample and time complexity. For simplicity, we assume $d_1 = d_2 = d$, $\kappa = \mathcal{O}(1)$.

*Remark 3* (Sample complexity and matrix completion). Using the lower bound on $p$, it is sufficient to have $\mathcal{O}(\mu^2 r^2 d \log d)$ observed entries. In the special case $S^* = 0$, our partial observation model is equivalent to the model of exact matrix completion (see, e.g., [8]). We note that our sample complexity (i.e., observations needed) matches that of completing a positive semidefinite (PSD) matrix by gradient descent as shown in [12], and is better than the non-convex matrix completion algorithms in [19] and [23]. Accordingly, our result reveals the important fact that we can obtain robustness in matrix completion without deterioration of our statistical guarantees. It is known that that any algorithm for solving exact matrix completion must have sample size $\Omega(\mu r d \log d)$ [8], and a nearly tight upper bound $O(\mu r d \log^2 d)$ is obtained in [10] by convex relaxation. While sub-optimal by a factor $\mu r$, our algorithm is much faster than convex relaxation as shown below.

*Remark 4* (Time complexity). Our sparse estimator on the sparse matrix with support $\Phi$ can be implemented via partial quick sort with running time $\mathcal{O}(pd^2 \log(\alpha pd))$. Computing the gradient in each step involves the two terms in the objective function (9). Computing the gradient of the first term $\widetilde{\mathcal{L}}$ takes time $\mathcal{O}(r|\Phi|)$, whereas the second term takes time $\mathcal{O}(r^2 d)$. In the initialization phase, performing rank-$r$ SVD on a sparse matrix with support $\Phi$ can be done in time $\mathcal{O}(r|\Phi|)$. We conclude that when $|\Phi| = \mathcal{O}(\mu^2 r^2 d \log d)$, Algorithm 2 achieves the error bound (15) with running time $\mathcal{O}(\mu^3 r^4 d \log d \log(1/\varepsilon))$. Therefore, in the small rank setting with $r \ll d^{1/3}$, even when full observations are given, it is better to use Algorithm 2 by subsampling the entries of $Y$.

## 5 Numerical Results

In this section, we provide numerical results and compare the proposed algorithms with existing methods, including the inexact augmented lagrange multiplier (IALM) approach [20] for solving the convex relaxation (1) and the alternating projection (AltProj) algorithm proposed in [21]. All algorithms are implemented in MATLAB [4], and the codes for existing algorithms are obtained from their authors. SVD computation in all algorithms uses the PROPACK library.[5] We ran all simulations on a machine with Intel 32-core Xeon (E5-2699) 2.3GHz with 240GB RAM.

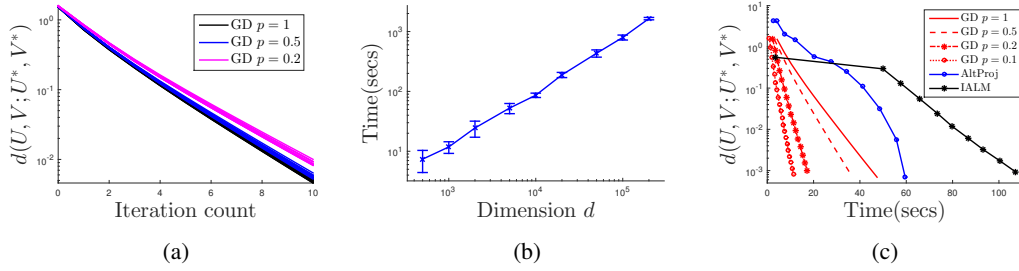

(a)&emsp;&emsp;&emsp;&emsp;&emsp;&emsp;&emsp;&emsp;(b)&emsp;&emsp;&emsp;&emsp;&emsp;&emsp;&emsp;&emsp;(c)

Figure 1: Results on synthetic data. (a) Plot of log estimation error versus number of iterations when using gradient descent (GD) with varying sub-sampling rate $p$. It is conducted using $d = 5000, r = 10, \alpha = 0.1$. (b) Plot of running time of GD versus dimension $d$ with $r = 10, \alpha = 0.1, p = 0.15r^2 \log d/d$. The low-rank matrix is recovered in all instances, and the line has slope approximately one. (c) Plot of log estimation error versus running time for different algorithms in problem with $d = 5000, r = 10, \alpha = 0.1$.

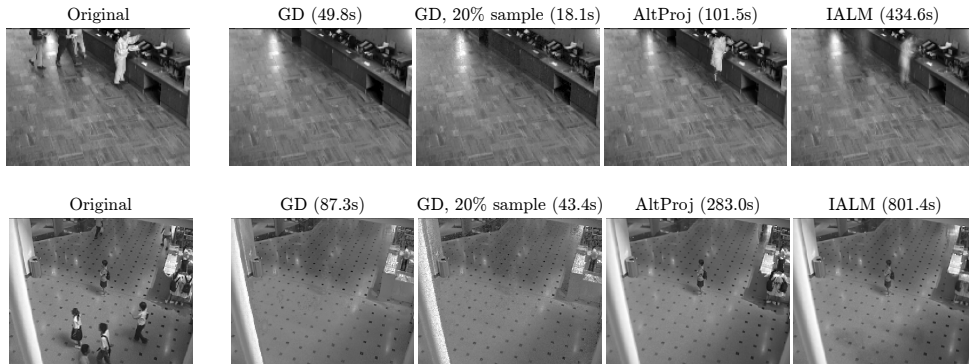

Figure 2: Foreground-background separation in *Restaurant* and *ShoppingMall* videos. In each line, the leftmost image is an original frame, and the other four are the separated background obtained from our algorithms with $p = 1$, $p = 0.2$, AltProj, and IALM. The running time required by each algorithm is shown in the title.

**Synthetic Datasets.** We generate a squared data matrix $Y = M^* + S^* \in \mathbb{R}^{d \times d}$ as follows. The low-rank part $M^*$ is given by $M^* = AB^\top$, where $A, B \in \mathbb{R}^{d \times r}$ have entries drawn independently from a zero mean Gaussian distribution with variance $1/d$. For a given sparsity parameter $\alpha$, each entry of $S^*$ is set to be nonzero with probability $\alpha$, and the values of the nonzero entries are sampled uniformly from $[-5r/d, 5r/d]$. The results are summarized in Figure 1. Figure 1a shows the convergence of our algorithms for different random instances with different sub-sampling rate $p$. Figure 1b shows the running time of our algorithm with partially observed data. We note that our algorithm is memory-efficient: in the large scale setting with $d = 2 \times 10^5$, using approximately 0.1% entries is sufficient for the successful recovery. In contrast, AltProj and IALM are designed to manipulate the entire matrix with $d^2 = 4 \times 10^{10}$ entries, which is prohibitive on single machine. Figure 1c compares our algorithms with AltProj and IALM by showing reconstruction error versus real running time. Our algorithm requires significantly less computation to achieve the same accuracy level, and using only a subset of the entries provides additional speed-up.

**Foreground-background Separation.** We apply our method to the task of foreground-background (FB) separation in a video. We use two public benchmarks, the *Restaurant* and *ShoppingMall* datasets.[6] Each dataset contains a video with static background. By vectorizing and stacking the frames as columns of a matrix $Y$, the FB separation problem can be cast as RPCA, where the static background corresponds to a low rank matrix $M^*$ with identical columns, and the moving objects in the video can be modeled as sparse corruptions $S^*$. Figure 2 shows the output of different algorithms on two frames from the dataset. Our algorithms require significantly less running time than both AltProj and IALM. Moreover, even with 20% sub-sampling, our methods still seem to achieve better separation quality. The details about parameter setting and more results are deferred to the supplemental material.

## Footnotes

[1]To ease presentation, the discussion here assumes $M^*$ has constant condition number, whereas our results below show the dependence on condition number explicitly.

[2] $\text{SVD}_r[A]$ stands for computing a rank-$r$ SVD of matrix $A$, i.e., finding the top $r$ singular values and vectors of $A$. Note that we only need to compute rank-$r$ SVD approximately (see the initialization error requirement in Theorem 1) so that we can leverage fast iterative approaches such as block power method and Krylov subspace methods.

[3]In fact, it suffices to compute the best rank-$r$ approximation with running time independent of the eigen gap.

[4]Our code is available at `https://www.yixinyang.org/code/RPCA_GD.zip`.

[5]`http://sun.stanford.edu/~rmunk/PROPACK/`

[6]http://perception.i2r.a-star.edu.sg/bk_model/bk_index.html

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
