[Supplementary Material]

# Fast RPCA: Supplemental Material

In this document, we collect all the proofs and technical lemmas for the theoretical results established in the main paper.

## A    Proofs

In this section we provide the proofs for our main theoretical results in Theorems 1–4 and Corollaries 1–2.

### A.1    Proof of Theorem 1

Let $\bar{Y} := Y - S_{\text{init}}$. As $Y = M^* + S^*$, we have $\bar{Y} - M^* = S^* - S_{\text{init}}$. We obtain $\bar{Y} - M^* \in \mathcal{S}_{2\alpha}$ because $S^*, S_{\text{init}} \in \mathcal{S}_\alpha$.

We claim that $\|\bar{Y} - M^*\|_\infty \leq 2\|M^*\|_\infty$. Denote the support of $S^*, S_{\text{init}}$ by $\Omega^*$ and $\Omega$ respectively. Since $\bar{Y} - M^*$ is supported on $\Omega \cup \Omega^*$, to prove the claim it suffices to consider the following three cases.

- For $(i, j) \in \Omega^* \cap \Omega$, due to rule of sparse estimation, we have $(S^* - S_{\text{init}})_{(i,j)} = 0$.

- For $(i, j) \in \Omega^* \setminus \Omega$, we must have $|S^*_{(i,j)}| \leq 2\|M^*\|_\infty$. Otherwise, we have $|Y_{(i,j)}| = |(S^* + M^*)_{(i,j)}| > \|M^*\|_\infty$. So $|Y_{(i,j)}|$ is larger than any uncorrupted entries in its row and column. Since there are at most $\alpha$ fraction corruptions per row and column, we have $Y_{(i,j)} \in \Omega$, which violates the prior condition $(i, j) \in \Omega^* \setminus \Omega$.

- For the last case $(i, j) \in \Omega \setminus \Omega^*$, since $(S_{\text{init}})_{(i,j)} = M^*_{(i,j)}$, trivially we have $|(S_{\text{init}})_{(i,j)}| \leq \|M^*\|_\infty$.

The following result, proved in Section B.1, relates the operator norm of $\bar{Y} - M^*$ to its infinite norm.

**Lemma 1.** *For any matrix $A \in \mathbb{R}^{d_1 \times d_2}$ that belongs to $\mathcal{S}_\alpha$ given in (3), we have*

$$\|A\|_{op} \leq \alpha\sqrt{d_1 d_2}\|A\|_\infty.$$

We thus obtain

$$\|\bar{Y} - M^*\|_{\text{op}} \leq 2\alpha\sqrt{d_1 d_2}\|\bar{Y} - M^*\|_\infty \leq 4\alpha\sqrt{d_1 d_2}\|M^*\|_\infty = 4\alpha\mu r\sigma_1^*. \tag{16}$$

In the last step, we use the fact that $M^*$ satisfies the $\mu$-incoherent condition, which leads to

$$\|M^*\|_\infty \leq \|M^*\|_{\text{op}}\|L^*\|_{2,\infty}\|R^*\|_{2,\infty} \leq \frac{\mu r}{\sqrt{d_1 d_2}}\|M^*\|_{\text{op}}. \tag{17}$$

We denote the $i$-th largest singular value of $\bar{Y}$ by $\sigma_i$. By Weyl's theorem, we have $|\sigma_i^* - \sigma_i| \leq \|\bar{Y} - M^*\|_{\text{op}}$ for all $i \in [d_1 \wedge d_2]$. Since $\sigma_{r+1}^* = 0$, we have $\sigma_{r+1} \leq \|\bar{Y} - M^*\|_{\text{op}}$. Recall that $U_0 V_0^\top$ is the best rank $r$ approximation of $\bar{Y}$. Accordingly, we have

$$\|U_0 V_0^\top - M^*\|_{\text{op}} \leq \|U_0 V_0^\top - \bar{Y}\|_{\text{op}} + \|\bar{Y} - M^*\|_{\text{op}}$$
$$= \sigma_{r+1} + \|\bar{Y} - M^*\|_{\text{op}} \leq 2\|\bar{Y} - M^*\|_{\text{op}} \leq 8\alpha\mu r\sigma_1^*.$$

Under condition $\alpha\mu r \leq \frac{1}{16\kappa}$, we obtain $\|U_0 V_0^\top - M^*\|_{\text{op}} \leq \frac{1}{2}\sigma_r^*$. Applying Lemma 5.14 in [24] (we provide it as Lemma 15 for the sake of completeness), we obtain

$$d^2(U_0, V_0; U^*, V^*) \leq \frac{2}{\sqrt{2}-1}\frac{\|U_0 V_0^\top - M^*\|_F^2}{\sigma_r^*} \leq \frac{10r\|U_0 V_0^\top - M^*\|_{\text{op}}^2}{\sigma_r^*}.$$

Plugging the upper bound of $\|U_0 V_0^\top - M^*\|_{\text{op}}$ into the above inequality completes the proof.

## A.2 Proof of Theorem 2

We essentially follow the general framework developed in [12] for analyzing the behaviors of gradient descent in factorized low-rank optimization. But it is worth to note that [12] only studies the symmetric and positive semidefinite setting, while we avoid such constraint on $M^*$. The techniques for analyzing general asymmetric matrix in factorized space is inspired by the recent work [24] on solving low-rank matrix equations. In our setting, the technical challenge is to verify the local descent condition of the loss function (8), which not only has a bilinear dependence on $U$ and $V$, but also involves our sparse estimator (4).

We begin with some notations. Define the equivalent set of optimal solution as

$$\mathcal{E}(M^*) := \left\{ (A, B) \in \mathbb{R}^{d_1 \times r} \times \mathbb{R}^{d_2 \times r} \mid A = L^* \Sigma^{*1/2} Q, B = R^* \Sigma^{*1/2} Q, \text{ where } Q \in \mathbb{Q}_r \right\}. \tag{18}$$

Given $(U_0, V_0) \in \mathbb{B}_2\left(c_2 \sqrt{\sigma_r^*/\kappa}\right)$, by (11), we have $\|U_0 V_0^\top - M^*\|_{\mathrm{op}} \leq \frac{1}{2}\sigma_r^*$ when $c_2$ is sufficiently small. By Weyl's theorem We thus have

$$\sqrt{\sigma_1^*/2} \leq \|U_0\|_{\mathrm{op}} \leq \sqrt{3\sigma_1^*/2}, \text{ and } \sqrt{\sigma_1^*/2} \leq \|V_0\|_{\mathrm{op}} \leq \sqrt{3\sigma_1^*/2}.$$

As a result, for $\mathcal{U}, \mathcal{V}$ constructed according to (7), we have

$$\mathcal{E}(M^*) \subseteq \mathcal{U} \times \mathcal{V}, \text{ and } \mathcal{U} \subseteq \bar{\mathcal{U}}, \mathcal{V} \subseteq \bar{\mathcal{V}}, \tag{19}$$

where

$$\bar{\mathcal{U}} := \left\{ A \in \mathbb{R}^{d_1 \times r} \mid \|A\|_{2,\infty} \leq \sqrt{\frac{3\mu r \sigma_1^*}{d_1}} \right\}, \quad \bar{\mathcal{V}} := \left\{ A \in \mathbb{R}^{d_2 \times r} \mid \|A\|_{2,\infty} \leq \sqrt{\frac{3\mu r \sigma_1^*}{d_2}} \right\}.$$

We let

$$\mathcal{G}(U, V) := \frac{1}{8}\|U^\top U - V^\top V\|_{\mathrm{F}}^2. \tag{20}$$

For $\mathcal{L}(U, V; S)$, we denote the gradient with respect to $M$ by $\nabla_M \mathcal{L}(U, V; S)$, i.e. $\nabla_M \mathcal{L}(U, V; S) = UV^\top + S - Y$.

The local descent property is implied by combining the following two results, which are proved in Section A.7 and A.8 respectively.

**Lemma 2** (Local descent property of $\mathcal{L}$). *Suppose $\mathcal{U}, \mathcal{V}$ satisfy (19). For any $(U, V) \in (\mathcal{U} \times \mathcal{V}) \cap \mathbb{B}_2(\sqrt{\sigma_1^*})$, we let $S = \mathcal{T}_{\gamma\alpha}\left[Y - UV^\top\right]$, where we choose $\gamma = 2$. Then we have that for $(U_{\pi^*}, V_{\pi^*}) \in \mathrm{argmin}_{(A,B) \in \mathcal{E}(M^*)} \|U - A\|_F^2 + \|V - B\|_F^2$ and $\beta > 0$,*

$$\langle\!\langle \nabla_M \mathcal{L}(U, V; S), UV^\top - U_{\pi^*} V_{\pi^*}^\top + \Delta_U \Delta_V^\top \rangle\!\rangle \geq \|UV^\top - U_{\pi^*} V_{\pi^*}^\top\|_F^2 - \nu \sigma_1^* \delta - 3\sqrt{\sigma_1^* \delta^3}. \tag{21}$$

*Here $\Delta_U := U - U_{\pi^*}$, $\Delta_V := V - V_{\pi^*}$, $\delta := \|\Delta_U\|_F^2 + \|\Delta_V\|_F^2$, and $\nu := 9(\beta + 6)\alpha\mu r + 5\beta^{-1}$.*

**Lemma 3** (Local descent property of $\mathcal{G}$). *For any $(U, V) \in \mathbb{B}_2(\sqrt{\sigma_r^*})$ and*

$$(U_{\pi^*}, V_{\pi^*}) \in \arg\min_{(A,B) \in \mathcal{E}(M^*)} \|U - A\|_F^2 + \|V - B\|_F^2,$$

*we have*

$$\langle\!\langle \nabla_U \mathcal{G}(U, V), U - U_{\pi^*} \rangle\!\rangle + \langle\!\langle \nabla_V \mathcal{G}(U, V), V - V_{\pi^*} \rangle\!\rangle$$

$$\geq \frac{1}{8}\|U^\top U - V^\top V\|_F^2 + \frac{1}{8}\sigma_r^* \delta - \sqrt{\frac{\sigma_1^* \delta^3}{2}} - \frac{1}{2}\|UV^\top - U_{\pi^*} V_{\pi^*}^\top\|_F^2,$$

*where $\delta$ is defined according to Lemma 2.*

As another key ingredient, we establish the following smoothness condition, proved in Section A.9, which indicates that the Frobenius norm of gradient decreases as $(U, V)$ approaches the optimal manifold.

**Lemma 4** (Smoothness). *For any $(U,V) \in \mathbb{B}_2(\sqrt{\sigma_1^*})$, we let $S = \mathcal{T}_{\gamma\alpha}\left[Y - UV^\top\right]$, where we choose $\gamma = 2$. We have that*

$$\|\nabla_M \mathcal{L}(U,V;S)\|_F^2 \le 6\|UV^\top - M^*\|_F^2, \tag{22}$$

*and*

$$\|\nabla_U \mathcal{G}(U,V)\|_F^2 + \|\nabla_V \mathcal{G}(U,V)\|_F^2 \le 2\sigma_1^*\|U^\top U - V^\top V\|_F^2. \tag{23}$$

With the above results in hand, we are ready to prove Theorem 2.

*Proof of Theorem 2.* We use shorthands

$$\delta_t := d^2(U_t, V_t; U^*, V^*), \ \mathcal{L}_t := \mathcal{L}(U_t, V_t; S_t), \ \text{and} \ \mathcal{G}_t := \mathcal{G}(U_t, V_t).$$

For $(U_t, V_t)$, let $(U_{\pi^*}^t, V_{\pi^*}^t) := \operatorname{argmin}_{(A,B)\in\mathcal{E}(M^*)} \|U_t - A\|_F^2 + \|V_t - B\|_F^2$. Define $\Delta_U^t := U_t - U_{\pi^*}^t$, $\Delta_V^t := V_t - V_{\pi^*}^t$.

We prove Theorem 2 by induction. It is sufficient to consider one step of the iteration. For any $t \ge 0$, under the induction hypothesis $(U_t, V_t) \in \mathbb{B}_2\left(c_2\sqrt{\sigma_r^*/\kappa}\right)$. We find that

$$
\begin{aligned}
\delta_{t+1} &\le \|U_{t+1} - U_{\pi^*}^t\|_F^2 + \|V_{t+1} - V_{\pi^*}^t\|_F^2 \\
&\le \|U_t - \eta\nabla_U\mathcal{L}_t - \eta\nabla_U\mathcal{G}_t - U_{\pi^*}^t\|_F^2 + \|V_t - \eta\nabla_V\mathcal{L}_t - \eta\nabla_V\mathcal{G}_t - V_{\pi^*}^t\|_F^2 \\
&\le \delta_t - 2\eta\underbrace{\langle\!\langle\nabla_U\mathcal{L}_t + \nabla_U\mathcal{G}_t,\ U_t - U_{\pi^*}^t\rangle\!\rangle}_{W_1} - 2\eta\underbrace{\langle\!\langle\nabla_V\mathcal{L}_t + \nabla_V\mathcal{G}_t,\ V_t - V_{\pi^*}^t\rangle\!\rangle}_{W_2} \\
&\quad + \eta^2\underbrace{\|\nabla_U\mathcal{L}_t + \nabla_U\mathcal{G}_t\|_F^2}_{W_3} + \eta^2\underbrace{\|\nabla_V\mathcal{L}_t + \nabla_V\mathcal{G}_t\|_F^2}_{W_4},
\end{aligned}
\tag{24}
$$

where the second step follows from the non-expansion property of projection onto $\mathcal{U}, \mathcal{V}$, which is implied by $\mathcal{E}(M^*) \subseteq \mathcal{U} \times \mathcal{V}$ shown in (19). Since $\nabla_U\mathcal{L}_t = [\nabla_M\mathcal{L}_t]V$ and $\nabla_V\mathcal{L}_t = [\nabla_M\mathcal{L}_t]^\top U$, we have

$$\langle\!\langle\nabla_U\mathcal{L}_t,\ U_t - U_{\pi^*}^t\rangle\!\rangle + \langle\!\langle\nabla_V\mathcal{L}_t,\ V_t - V_{\pi^*}^t\rangle\!\rangle = \langle\!\langle\nabla_M\mathcal{L}_t,\ U_t V_t^\top - U_{\pi^*}^t V_{\pi^*}^{t\top} + \Delta_U^t \Delta_V^{t\top}\rangle\!\rangle.$$

Combining Lemma 2 and 3, under condition $\delta_t < \sigma_r^*$, we have that

$$W_1 + W_2 \ge \frac{1}{2}\|U_t V_t^\top - M^*\|_F^2 + \frac{1}{8}\|U_t^\top U_t - V_t^\top V_t\|_F^2 + \frac{1}{8}\sigma_r^*\delta_t - \nu\sigma_1^*\delta_t - 4\sqrt{\sigma_1^*\delta_t^3}.$$

On the other hand, we have

$$
\begin{aligned}
W_3 + W_4 &\le 2\|\nabla_U\mathcal{L}_t\|_F^2 + 2\|\nabla_U\mathcal{G}_t\|_F^2 + 2\|\nabla_V\mathcal{L}_t\|_F^2 + 2\|\nabla_V\mathcal{G}_t\|_F^2 \\
&\le 2(\|U_t\|_{\text{op}}^2 + \|V_t\|_{\text{op}}^2)\|\nabla_M\mathcal{L}_t\|_F^2 + 2\|\nabla_U\mathcal{G}_t\|_F^2 + 2\|\nabla_V\mathcal{G}_t\|_F^2 \\
&\le 36\sigma_1^*\|U_t V_t^\top - M^*\|_F^2 + 4\sigma_1^*\|U_t^\top U_t - V_t^\top V_t\|_F^2,
\end{aligned}
$$

where the last step is implied by Lemma 4 and the assumption $(U_t, V_t) \in \mathbb{B}_2\left(c_2\sqrt{\sigma_r^*/\kappa}\right)$ that leads to $\|U_t\|_{\text{op}} \le \sqrt{3\sigma_1^*/2}$, $\|V_t\|_{\text{op}} \le \sqrt{3\sigma_1^*/2}$.

By the assumption $\eta = c/\sigma_1^*$ for any constant $c \le 1/36$, we thus have

$$-2\eta(W_1 + W_2) + \eta^2(W_3 + W_4) \le -\frac{1}{4}\eta\sigma_r^*\delta_t + 2\eta\nu\sigma_1^*\delta_t + 8\eta\sqrt{\sigma_1^*\delta_t^3}.$$

In Lemma 2, choosing $\beta = 320\kappa$ and assuming $\alpha \lesssim 1/(\kappa^2\mu r)$, we can have $\nu \le 1/(32\kappa)$. Assuming $\delta_t \lesssim \sigma_r^*/\kappa$ leads to $14\sqrt{\sigma_1^*\delta_t^3} \le \frac{1}{16}\sigma_r^*\delta_t$. We thus obtain

$$\delta_{t+1} \le \left(1 - \frac{\eta\sigma_r^*}{8}\right)\delta_t. \tag{25}$$

Under initial condition $\delta_0 \lesssim \sigma_r^*/\kappa$, we obtain that such condition holds for all $t$ since estimation error decays geometrically after each iteration. Then applying (25) for all iterations, we conclude that for all $t = 0, 1, \ldots$,

$$\delta_t \le \left(1 - \frac{\eta\sigma_r^*}{8}\right)^t \delta_0.$$

$\square$

## A.3 Proof of Corollary 1

We need $\alpha \lesssim \frac{1}{\kappa^2 \mu r}$ due to the condition of Theorem 2. In order to ensure the linear convergence happens, it suffices to let the initial error shown in Theorem 1 be less than the corresponding condition in Theorem 2. Accordingly, we need

$$28\sqrt{\kappa}\alpha\mu r\sqrt{r}\sqrt{\sigma_1^*} \lesssim \sqrt{\sigma_r^*/\kappa},$$

which leads to $\alpha \lesssim \frac{1}{\mu\sqrt{r\kappa^3}}$.

Using the conclusion that gradient descent has linear convergence, choosing $T = \mathcal{O}(\kappa \log(1/\varepsilon))$, we have

$$d^2(U_T, V_T; U^*, V^*) \le \varepsilon^2 d^2(U_0, V_0; U^*, V^*) \lesssim \varepsilon^2 \frac{\sigma_r^*}{\kappa}.$$

Finally, applying the relationship between $d(U_T, V_T; U^*, V^*)$ and $\|U_T V_T^\top - M^*\|_{\mathrm{F}}$ shown in (11), we complete the proof.

## A.4 Proof of Theorem 3

Let $\bar{Y} := \frac{1}{p}(Y - S_{\text{init}})$. Similar to the proof of Theorem 1, we first establish an upper bound on $\|\bar{Y} - M^*\|_{\text{op}}$. We have that

$$\|\bar{Y} - M^*\|_{\text{op}} \le \|\bar{Y} - \frac{1}{p}\Pi_\Phi M^*\|_{\text{op}} + \|\frac{1}{p}\Pi_\Phi(M^*) - M^*\|_{\text{op}}. \tag{26}$$

For the first term, we have $\bar{Y} - \frac{1}{p}\Pi_\Phi M^* = \frac{1}{p}(\Pi_\Phi(S^*) - S_{\text{init}})$ because $Y = \Pi_\Phi(M^* + S^*)$. Lemma 10 shows that under condition $p \gtrsim \frac{\log d}{\alpha(d_1 \wedge d_2)}$, there are at most $\frac{3}{2}p\alpha$-fraction nonzero entries in each row and column of $\Pi_\Phi(S^*)$ with high probability. Since $S_{\text{init}} \in \mathcal{S}_{2p\alpha}$, we have

$$\Pi_\Phi(S^*) - S_{\text{init}} \in \mathcal{S}_{4p\alpha}. \tag{27}$$

In addition, we prove below that

$$\|\Pi_\Phi(S^*) - S_{\text{init}}\|_\infty \le 2\|M^*\|_\infty. \tag{28}$$

Denote the support of $\Pi_\Phi(S^*)$ and $S_{\text{init}}$ by $\Omega_o^*$ and $\Omega$. For $(i,j) \in \Omega_o^* \cap \Omega$ and $(i,j) \in \Omega \setminus \Omega_o^*$, we have $(\Pi_\Phi(S^*) - S_{\text{init}})_{(i,j)} = 0$ and $(\Pi_\Phi(S^*) - S_{\text{init}})_{(i,j)} = -M^*_{(i,j)}$, respectively. To prove the claim, it remains to show that for $(i,j) \in \Omega_o^* \setminus \Omega$, $|S^*_{(i,j)}| < 2\|M^*\|_\infty$. If this is not true, then we must have $|Y_{(i,j)}| > \|M^*\|_\infty$. Accordingly, $|Y_{(i,j)}|$ is larger than the magnitude of any uncorrupted entries in its row and column. Note that on the support $\Phi$, there are at most $\frac{3}{2}p\alpha$ corruptions per row and column, we have $(i,j) \in \Omega$, which violates our prior condition $(i,j) \in \Omega_o^* \setminus \Omega$.

Using these two properties (27), (28) and applying Lemma 1, we have

$$\|\bar{Y} - \frac{1}{p}\Pi_\Phi M^*\|_{\text{op}} \le 4\alpha\sqrt{d_1 d_2}\|\Pi_\Phi(S^*) - S_{\text{init}}\|_\infty \le 8\alpha\sqrt{d_1 d_2}\|M^*\|_\infty \le 8\alpha\mu r\sigma_1^*, \tag{29}$$

where the last step follow from (17).

For the second term in (26), we use the following lemma proved in [10].

**Lemma 5** (Lemma 2 in [10]). *Suppose* $A \in \mathbb{R}^{d_1 \times d_2}$ *is a fixed matrix. We let* $d := \max\{d_1, d_2\}$. *There exists a constant* $c$ *such that with probability at least* $1 - \mathcal{O}(d^{-1})$,

$$\|\frac{1}{p}\Pi_\Phi(A) - A\|_{op} \le c\left(\frac{\log d}{p}\|A\|_\infty + \sqrt{\frac{\log d}{p}}\max\left\{\|A\|_{2,\infty}, \|A^\top\|_{2,\infty}\right\}\right).$$

Given the SVD $M^* = L^*\Sigma R^{*\top}$, for any $i \in [d_1]$, we have

$$\|M^*_{(i,\cdot)}\|_2 = \|L^*_{(i,\cdot)}\Sigma R^{*\top}\|_2 \le \sigma_1^*\|L^*_{(i,\cdot)}\|_2 \le \sigma_1^*\sqrt{\frac{\mu r}{d_1}}.$$

We can bound $\|M^{*\top}\|_{2,\infty}$ similarly. Lemma 5 leads to

$$\left\|\frac{1}{p}\Pi_\Phi(M^*) - M^*\right\|_{\mathrm{op}} \leq c'\frac{\log d}{p}\frac{\mu r}{\sqrt{d_1 d_2}}\sigma_1^* + c'\sqrt{\frac{\log d}{p}}\sqrt{\frac{\mu r}{d_1 \wedge d_2}}\sigma_1^* \leq c'\epsilon\sigma_1^*/\sqrt{r} \qquad (30)$$

under condition $p \geq \frac{4\mu r^2 \log d}{\epsilon^2 (d_1 \wedge d_2)}$.

Putting (29) and (30) together, we obtain

$$\|\bar{Y} - M^*\|_{\mathrm{op}} \leq 8\alpha\mu r\sigma_1^* + c'\epsilon\sigma_1^*/\sqrt{r}.$$

Then using the fact that $U_0 V_0^\top$ is the best rank $r$ approximation of $\bar{Y}$ and applying Wely's theorem (see the proof of Theorem 1 for a detailed argument), we have

$$\|U_0 V_0^\top - M^*\|_{\mathrm{op}} \leq \|U_0 V_0^\top - \bar{Y}\|_{\mathrm{op}} + \|\bar{Y} - M^*\|_{\mathrm{op}}$$
$$\leq 2\|\bar{Y} - M^*\|_{\mathrm{op}} \leq 16\alpha\mu r\sigma_1^* + 2c'\epsilon\sigma_1^*/\sqrt{r}$$

Under our assumptions, we have $16\alpha\mu r\sigma_1^* + 2c'\epsilon\sigma_1^*/\sqrt{r} \leq \frac{1}{2}\sigma_r^*$. Accordingly, Lemma 15 gives

$$d^2(U_0, V_0; U^*, V^*) \leq \frac{2}{\sqrt{2}-1}\frac{\|U_0 V_0^\top - M^*\|_F^2}{\sigma_r^*} \leq \frac{10r\|U_0 V_0^\top - M^*\|_{\mathrm{op}}^2}{\sigma_r^*}.$$

We complete the proof by combining the above two inequalities.

## A.5 Proof of Theorem 4

In this section, we turn to prove Theorem 4. Similar to the proof of Theorem 2, we rely on establishing the local descent and smoothness conditions. Compared to the full observation setting, we replace $\mathcal{L}$ by $\widetilde{\mathcal{L}}$ given in (6), while the regularization term $\widetilde{\mathcal{G}}(U,V) := \frac{1}{64}\|U^\top U - V^\top V\|_F^2$ merely differs from $\mathcal{G}(U,V)$ given in (20) by a constant factor. It is thus sufficient to analyze the properties of $\widetilde{\mathcal{L}}$.

Define $\mathcal{E}(M^*)$ according to (18). Under the initial condition, we still have

$$\mathcal{E}(M^*) \subseteq \mathcal{U} \times \mathcal{V}, \text{ and } \mathcal{U} \subseteq \bar{\mathcal{U}}, \mathcal{V} \subseteq \bar{\mathcal{V}}. \qquad (31)$$

We prove the next two lemmas in Section A.10 and A.11 respectively. In both lemmas, for any $(U, V) \in \mathcal{U} \times \mathcal{V}$, we use shorthands

$$(U_{\pi^*}, V_{\pi^*}) = \arg\min_{(A,B)\in\mathcal{E}(M^*)} \|U - A\|_F^2 + \|V - B\|_F^2,$$

$\Delta_U := U - U_{\pi^*}$, $\Delta_V := V - V_{\pi^*}$, and $\delta := \|\Delta_U\|_F^2 + \|\Delta_V\|_F^2$. Recall that $d := \max\{d_1, d_2\}$.

**Lemma 6** (Local descent property of $\widetilde{\mathcal{L}}$)**.** *Suppose* $\mathcal{U}, \mathcal{V}$ *satisfy* (31). *Suppose we let*

$$S = \mathcal{T}_{\gamma p\alpha}\left[\Pi_\Phi\left(Y - UV^\top\right)\right],$$

*where we choose* $\gamma = 3$. *For any* $\beta > 0$ *and* $\epsilon \in (0, \frac{1}{4})$, *we define* $\nu := (14\beta + 81)\alpha\mu r + 26\sqrt{\epsilon} + 18\beta^{-1}$. *There exist constants* $\{c_i\}_{i=1}^2$ *such that if*

$$p \geq c_1\left(\frac{\mu^2 r^2}{\epsilon^2} + \frac{1}{\alpha} + 1\right)\frac{\log d}{d_1 \wedge d_2}, \qquad (32)$$

*then with probability at least* $1 - c_2 d^{-1}$,

$$\langle\!\langle \nabla_M\widetilde{\mathcal{L}}(U, V; S), UV^\top - U_{\pi^*}V_{\pi^*}^\top + \Delta_U\Delta_V^\top\rangle\!\rangle \geq \frac{3}{16}\|UV^\top - U_{\pi^*}V_{\pi^*}^\top\|_F^2 - \nu\sigma_1^*\delta - 10\sqrt{\sigma_1^{*3}\delta^3} - 2\delta^2 \qquad (33)$$

*for all* $(U, V) \in (\mathcal{U} \times \mathcal{V}) \cap \mathbb{B}_2\left(\sqrt{\sigma_1^*}\right)$.

**Lemma 7** (Smoothness of $\widetilde{\mathcal{L}}$)**.** *Suppose* $\mathcal{U}, \mathcal{V}$ *satisfy* (31). *Suppose we let* $S = \mathcal{T}_{\gamma\alpha p}\left[\Pi_\Phi\left(Y - UV^\top\right)\right]$ *for* $\gamma = 3$. *There exist constants* $\{c_i\}_{i=1}^3$ *such that for any* $\epsilon \in (0, \frac{1}{4})$, *when* $p$ *satisfies condition* (32), *with probability at least* $1 - c_2 d^{-1}$, *we have that for all* $(U, V) \in (\mathcal{U} \times \mathcal{V}) \cap \mathbb{B}_2(\sqrt{\sigma_1^*})$,

$$\|\nabla_U\widetilde{\mathcal{L}}(U, V; S)\|_F^2 + \|\nabla_V\widetilde{\mathcal{L}}(U, V; S)\|_F^2 \leq c_3\left[\mu r\sigma_1^*\|UV^\top - U_{\pi^*}V_{\pi^*}^\top\|_F^2 + \mu r\sigma_1^*\delta(\delta + \epsilon\sigma_1^*)\right]. \qquad (34)$$

In the remainder of this section, we condition on the events in Lemma 6 and 7. Now we are ready to prove Theorem 4.

*Proof of Theorem 4.* We essentially follow the process for proving Theorem 2. Let the following shorthands be defined in the same fashion: $\delta_t$, $(U_{\pi^*}^t, V_{\pi^*}^t)$, $(\Delta_U^t, \Delta_V^t)$, $\widetilde{\mathcal{L}}_t$, $\widetilde{\mathcal{G}}_t$.

Here we show error decays in one step of iteration. The induction process is the same as the proof of Theorem 2, and is thus omitted. For any $t \geq 0$, similar to (24) we have that

$$\delta_{t+1} \leq \delta_t - 2\eta \underbrace{\langle\!\langle \nabla_U \widetilde{\mathcal{L}}_t + \nabla_U \widetilde{\mathcal{G}}_t,\ U_t - U_{\pi^*}^t \rangle\!\rangle}_{W_1} - 2\eta \underbrace{\langle\!\langle \nabla_V \widetilde{\mathcal{L}}_t + \nabla_V \widetilde{\mathcal{G}}_t,\ V_t - V_{\pi^*}^t \rangle\!\rangle}_{W_2}$$
$$+ \eta^2 \underbrace{\|\nabla_U \widetilde{\mathcal{L}}_t + \nabla_U \widetilde{\mathcal{G}}_t\|_F^2}_{W_3} + \eta^2 \underbrace{\|\nabla_V \widetilde{\mathcal{L}}_t + \nabla_V \widetilde{\mathcal{G}}_t\|_F^2}_{W_4}.$$

We also have

$$\langle\!\langle \nabla_U \widetilde{\mathcal{L}}_t,\ U_t - U_{\pi^*}^t \rangle\!\rangle + \langle\!\langle \nabla_V \widetilde{\mathcal{L}}_t,\ V_t - V_{\pi^*}^t \rangle\!\rangle = \langle\!\langle \nabla_M \widetilde{\mathcal{L}}_t,\ U_t V_t^\top - U_{\pi^*}^t V_{\pi^*}^{t\top} + \Delta_U^t \Delta_V^{t\top} \rangle\!\rangle,$$

which can be lower bounded by Lemma 6. Note that $\widetilde{\mathcal{G}}$ differs from $\mathcal{G}$ by a constant, we can still leverage Lemma 3. Hence, we obtain that

$$W_1 + W_2 \geq \frac{1}{8}\|U_t V_t^\top - M^*\|_F^2 + \frac{1}{64}\|U_t^\top U_t - V_t^\top V_t\|_F^2 + \frac{1}{64}\sigma_r^*\delta_t - \nu\sigma_1^*\delta_t - 11\sqrt{\sigma_1^*\delta_t^3} - 2\delta_t^2.$$

On the other hand, we have

$$W_3 + W_4 \leq 2\|\nabla_U \widetilde{\mathcal{L}}_t\|_F^2 + 2\|\nabla_U \widetilde{\mathcal{G}}_t\|_F^2 + 2\|\nabla_V \widetilde{\mathcal{L}}_t\|_F^2 + 2\|\nabla_V \widetilde{\mathcal{G}}_t\|_F^2$$
$$\leq c\left[\mu r\sigma_1^*\|U_t V_t^\top - M^*\|_F^2 + \mu r\sigma_1^*\delta_t(\delta_t + \epsilon\sigma_1^*) + \sigma_1^*\|U_t^\top U_t - V_t^\top V_t\|_F^2\right],$$

where $c$ is a constant, and the last step is implied by Lemma 4 and Lemma 7.

By the assumption $\eta = c'/[\mu r\sigma_1^*]$ for sufficiently small constant $c'$, we thus have

$$-2\eta(W_1 + W_2) + \eta^2(W_3 + W_4) \leq -\frac{1}{32}\eta\sigma_r^*\delta_t + 2\eta\nu\sigma_1^*\delta_t + 22\eta\sqrt{\sigma_1^*\delta_t^3} + 4\eta\delta_t^2.$$

Recall that $\nu := (14\beta + 81)\alpha\mu r + 26\sqrt{\epsilon} + 18\beta^{-1}$. By letting $\beta = c_1\kappa$, $\epsilon = c_2/\kappa^2$ and assuming $\alpha \leq c_3/(\mu r\kappa^2)$ and $\delta_t \leq c_4\sigma_r^*/\kappa$ for some sufficiently small constants $\{c_i\}_{i=1}^4$, we can have $-2\eta(W_1 + W_2) + \eta^2(W_3 + W_4) \leq -\frac{1}{64}\eta\sigma_r^*\delta_t$, which implies that

$$\delta_{t+1} \leq \left(1 - \frac{\eta\sigma_r^*}{64}\right)\delta_t,$$

and thus completes the proof. $\qquad\square$

## A.6 Proof of Corollary 2

We need $\alpha \lesssim \frac{1}{\mu\kappa^2 r}$ due to the condition of Theorem 4. Letting the initial error provided in Theorem 3 be less than the corresponding condition in Theorem 4, we have

$$51\sqrt{\kappa}\alpha\mu r\sqrt{r}\sqrt{\sigma_1^*} + 7c_1\epsilon\sqrt{\kappa\sigma_1^*} \lesssim \sqrt{\sigma_r^*/\kappa},$$

which leads to

$$\alpha \lesssim \frac{1}{\mu\sqrt{r^3\kappa^3}},\quad \epsilon \lesssim \frac{1}{\sqrt{\kappa^3}}.$$

Plugging the above two upper bounds into the second term in (13), it suffices to have

$$p \gtrsim \frac{\kappa^3\mu r^2 \log d}{d_1 \wedge d_2}.$$

Comparing the above bound with the second term in (14) completes the proof.

### A.7 Proof of Lemma 2

Let $M := UV^\top$. We observe that

$$\nabla_M \mathcal{L}(U, V; S) = M + S - M^* - S^*.$$

Plugging it back into the left hand side of (21), we obtain

$$\langle\!\langle \nabla_M \mathcal{L}(U, V; S), \ UV^\top - U_{\pi^*} V_{\pi^*}^\top + \Delta_U \Delta_V^\top \rangle\!\rangle = \langle\!\langle M + S - M^* - S^*, \ M - M^* + \Delta_U \Delta_V^\top \rangle\!\rangle$$
$$\geq \|M - M^*\|_F^2 - \underbrace{|\langle\!\langle S - S^*, \ M - M^* \rangle\!\rangle|}_{T_1} - \underbrace{|\langle\!\langle M + S - M^* - S^*, \ \Delta_U \Delta_V^\top \rangle\!\rangle|}_{T_2}. \tag{35}$$

Next we derive upper bounds of $T_1$ and $T_2$ respectively.

**Upper bound of $T_1$.** We denote the support of $S, S^*$ by $\Omega$ and $\Omega^*$ respectively. Since $S - S^*$ is supported on $\Omega^* \cup \Omega$, we have

$$T_1 \leq \underbrace{|\langle\!\langle \Pi_\Omega(S - S^*), \ M - M^* \rangle\!\rangle|}_{W_1} + \underbrace{|\langle\!\langle \Pi_{\Omega^* \setminus \Omega}(S - S^*), \ M - M^* \rangle\!\rangle|}_{W_2}.$$

Recall that for any $(i, j) \in \Omega$, we have $S_{(i,j)} = (M^* + S^* - M)_{(i,j)}$. Accordingly, we have

$$W_1 = \|\Pi_\Omega(M - M^*)\|_F^2. \tag{36}$$

Now we turn to bound $W_2$. Since $S_{(i,j)} = 0$ for any $(i, j) \in \Omega^* \setminus \Omega$, we have

$$W_2 = |\langle\!\langle \Pi_{\Omega^* \setminus \Omega} S^*, \ M - M^* \rangle\!\rangle|.$$

Let $u_i$ be the $i$-th row of $M - M^*$, and $v_j$ be the $j$-th column of $M - M^*$. For any $k \in [d_2]$, we let $u_i^{(k)}$ denote the element of $u_i$ that has the $k$-th largest magnitude. Similarly, for any $k \in [d_1]$, we let $v_j^{(k)}$ denote the element of $v_j$ that has the $k$-th largest magnitude.

From the design of sparse estimator (4), we have that for any $(i, j) \in \Omega^* \setminus \Omega$, $|(M^* + S^* - M)_{(i,j)}|$ is either smaller than the $\gamma \alpha d_2$-th largest entry of the $i$-th row of $M^* + S^* - M$ or smaller than the $\gamma \alpha d_1$-th largest entry of the $j$-th column of $M^* + S^* - M$. Note that $S^*$ only contains at most $\alpha$-fraction nonzero entries per row and column. As a result, $|(M^* + S^* - M)_{(i,j)}|$ has to be less than the magnitude of $u_i^{(\gamma \alpha d_2 - \alpha d_2)}$ or $v_j^{(\gamma \alpha d_1 - \alpha d_1)}$. Formally, we have for $(i, j) \in \Omega^* \setminus \Omega$,

$$|(M^* + S^* - M)_{(i,j)}| \leq \underbrace{\max\left\{ |u_i^{(\gamma \alpha d_2 - \alpha d_2)}|, \ |v_j^{(\gamma \alpha d_1 - \alpha d_1)}| \right\}}_{b_{ij}}. \tag{37}$$

Furthermore, we obtain

$$b_{ij}^2 \leq |u_i^{(\gamma \alpha d_2 - \alpha d_2)}|^2 + |v_j^{(\gamma \alpha d_1 - \alpha d_1)}|^2 \leq \frac{\|u_i\|_2^2}{(\gamma - 1)\alpha d_2} + \frac{\|v_j\|_2^2}{(\gamma - 1)\alpha d_1}. \tag{38}$$

Meanwhile, for any $(i, j) \in \Omega^* \setminus \Omega$, we have

$$|S^*_{(i,j)} \cdot (M - M^*)_{(i,j)}| = |(M^* + S^* - M - M^* + M)_{(i,j)} \cdot (M - M^*)_{(i,j)}|$$
$$\leq |(M - M^*)_{(i,j)}|^2 + |(M^* + S^* - M)_{(i,j)}| \cdot |(M - M^*)_{(i,j)}|$$
$$\leq |(M - M^*)_{(i,j)}|^2 + b_{ij} \cdot |(M - M^*)_{(i,j)}|$$
$$\leq \left(1 + \frac{\beta}{2}\right) |(M - M^*)_{(i,j)}|^2 + \frac{b_{ij}^2}{2\beta}, \tag{39}$$

where $\beta$ in the last step can be any positive number. Combining (38) and (39) leads to

$$W_2 \leq \sum_{(i,j) \in \Omega^* \setminus \Omega} |S^*_{(i,j)} \cdot (M - M^*)_{(i,j)}|$$

$$\leq \left(1 + \frac{\beta}{2}\right) \|\Pi_{\Omega^* \setminus \Omega}(M - M^*)\|_{\mathrm{F}}^2 + \sum_{(i,j) \in \Omega^* \setminus \Omega} \frac{b_{ij}^2}{2\beta}$$

$$\leq \left(1 + \frac{\beta}{2}\right) \|\Pi_{\Omega^* \setminus \Omega}(M - M^*)\|_{\mathrm{F}}^2 + \frac{1}{2\beta} \sum_{(i,j) \in \Omega^* \setminus \Omega} \left(\frac{\|u_i\|_2^2}{(\gamma - 1)\alpha d_2} + \frac{\|v_j\|_2^2}{(\gamma - 1)\alpha d_1}\right)$$

$$\leq \left(1 + \frac{\beta}{2}\right) \|\Pi_{\Omega^* \setminus \Omega}(M - M^*)\|_{\mathrm{F}}^2 + \frac{1}{\beta(\gamma - 1)} \|M - M^*\|_{\mathrm{F}}^2. \tag{40}$$

In the last step, we use

$$\sum_{(i,j) \in \Omega^* \setminus \Omega} \left(\frac{1}{d_2}\|u_i\|_2^2 + \frac{1}{d_1}\|v_j\|_2^2\right) \leq \sum_{(i,j) \in \Omega^*} \left(\frac{1}{d_2}\|u_i\|_2^2 + \frac{1}{d_1}\|v_j\|_2^2\right)$$

$$\leq \sum_{i \in [d]} \sum_{j \in \Omega^*_{(i,\cdot)}} \frac{1}{d_2}\|u_i\|_2^2 + \sum_{j \in [d]} \sum_{i \in \Omega^*_{(\cdot,j)}} \frac{1}{d_1}\|v_j\|_2^2$$

$$\leq \alpha \sum_{i \in [d]} \|u_i\|_2^2 + \alpha \sum_{j \in [d]} \|v_j\|_2^2 = 2\alpha \|M - M^*\|_{\mathrm{F}}^2. \tag{41}$$

We introduce shorthand $\delta := \|\Delta_U\|_{\mathrm{F}}^2 + \|\Delta_V\|_{\mathrm{F}}^2$. We prove the following inequality in the end of this section.

$$\|M - M^*\|_{\mathrm{F}} \leq \sqrt{5\sigma_1^* \delta}. \tag{42}$$

Combining (36), (40) and (42) leads to

$$T_1 \leq \|\Pi_\Omega(M - M^*)\|_{\mathrm{F}}^2 + \left(1 + \frac{\beta}{2}\right) \|\Pi_{\Omega^* \setminus \Omega}(M - M^*)\|_{\mathrm{F}}^2 + \frac{5\sigma_1^* \delta}{\beta(\gamma - 1)}$$

$$\leq 9(2\gamma + \beta + 2)\alpha\mu r \sigma_1^* \delta + \frac{5\sigma_1^* \delta}{\beta(\gamma - 1)}, \tag{43}$$

where the last step follows from Lemma 14 by noticing that $\Pi_\Omega(M - M^*)$ has at most $\gamma\alpha$-fraction nonzero entries per row and column.

**Upper bound of $T_2$.** To ease notation, we let $C := M + S - M^* - S^*$. We observe that $C$ is supported on $\Omega^c$, we have

$$T_2 \leq \underbrace{|\langle\!\langle \Pi_{\Omega^{*c} \cap \Omega^c}(M - M^*), \ \Delta_U \Delta_V^\top \rangle\!\rangle|}_{W_3} + \underbrace{|\langle\!\langle \Pi_{\Omega^* \cap \Omega^c} C, \ \Delta_U \Delta_V^\top \rangle\!\rangle|}_{W_4}.$$

By Cauchy-Swartz inequality, we have

$$W_3 \leq \|\Pi_{\Omega^{*c} \cap \Omega^c}(M - M^*)\|_{\mathrm{F}} \|\Delta_U \Delta_V^\top\|_{\mathrm{F}} \leq \|M - M^*\|_{\mathrm{F}} \|\Delta_U\|_{\mathrm{F}} \|\Delta_V\|_{\mathrm{F}} \leq \sqrt{5\sigma_1^* \delta^3}/2,$$

where the last step follows from (42) and $\|\Delta_U\|_{\mathrm{F}} \|\Delta_V\|_{\mathrm{F}} \leq \delta/2$.

It remains to bound $W_4$. By Cauchy-Swartz inequality, we have

$$W_4 \leq \|\Pi_{\Omega^* \cap \Omega^c} C\|_{\mathrm{F}} \|\Delta_U \Delta_V^\top\|_{\mathrm{F}} \leq \|\Pi_{\Omega^* \cap \Omega^c}(M^* + S^* - M)\|_{\mathrm{F}} \|\Delta_U \Delta_V^\top\|_{\mathrm{F}}$$

$$\overset{(a)}{\leq} \sqrt{\sum_{(i,j) \in \Omega^* \setminus \Omega} b_{ij}^2} \|\Delta_U\|_{\mathrm{F}} \|\Delta_V\|_{\mathrm{F}} \overset{(b)}{\leq} \left[\sum_{(i,j) \in \Omega^* \setminus \Omega} \frac{\|u_i\|_2^2}{(\gamma - 1)\alpha d_2} + \frac{\|v_j\|_2^2}{(\gamma - 1)\alpha d_1}\right]^{1/2} \|\Delta_U\|_{\mathrm{F}} \|\Delta_V\|_{\mathrm{F}}.$$

$$\overset{(c)}{\leq} \sqrt{\frac{2}{\gamma - 1}} \|M - M^*\|_{\mathrm{F}} \|\Delta_U\|_{\mathrm{F}} \|\Delta_V\|_{\mathrm{F}} \leq \sqrt{\frac{5\sigma_1^* \delta^3}{2(\gamma - 1)}},$$

where step $(a)$ is from (37), step $(b)$ follows from (38), and step $(c)$ follows from (41). Combining the upper bounds of $W_3$ and $W_4$, we obtain

$$T_2 \leq \sqrt{5\sigma_1^* \delta^3}/2 + \sqrt{\frac{5\sigma_1^* \delta^3}{2(\gamma - 1)}}. \tag{44}$$

**Combining pieces.** Now we choose $\gamma = 2$. Then inequality (43) implies that
$$T_1 \leq [9(\beta + 6)\alpha\mu r + 5\beta^{-1}]\sigma_1^* \delta.$$
Inequality (44) then implies that
$$T_2 \leq 3\sqrt{\sigma_1^* \delta^3}.$$
Plugging the above two inequalities into (35) completes the proof.

*Proof of inequality* (42). We find that

$$\|M - M^*\|_{\mathrm{F}}^2 \leq \left[\sqrt{\sigma_1^*}(\|\Delta_V\|_{\mathrm{F}} + \|\Delta_U\|_{\mathrm{F}}) + \|\Delta_U\|_{\mathrm{F}}\|\Delta_V\|_{\mathrm{F}}\right]^2$$

$$\leq \left[\sqrt{\sigma_1^*}(\|\Delta_V\|_{\mathrm{F}} + \|\Delta_U\|_{\mathrm{F}}) + \frac{1}{2}\sqrt{\sigma_1^*}\|\Delta_U\|_{\mathrm{F}} + \frac{1}{2}\sqrt{\sigma_1^*}\|\Delta_V\|_{\mathrm{F}}\right]^2$$

$$\leq 5\sigma_1^*(\|\Delta_U\|_{\mathrm{F}}^2 + \|\Delta_V\|_{\mathrm{F}}^2),$$

where the first step follows from the upper bound of $\|M - M^*\|_{\mathrm{F}}$ shown in Lemma 12, and the second step follows from the assumption $\|\Delta_U\|_{\mathrm{F}}, \|\Delta_V\|_{\mathrm{F}} \leq \sqrt{\sigma_1^*}$.

## A.8 Proof of Lemma 3

We first observe that
$$\nabla_U \mathcal{G}(U,V) = \frac{1}{2}U(U^\top U - V^\top V), \ \nabla_V \mathcal{G}(U,V) = \frac{1}{2}V(V^\top V - U^\top U),$$
Therefore, we obtain

$$\langle\!\langle \nabla_U \mathcal{G}(U,V), \ U - U_{\pi^*} \rangle\!\rangle + \langle\!\langle \nabla_V \mathcal{G}(U,V), \ V - V_{\pi^*} \rangle\!\rangle$$

$$= \frac{1}{2} \langle\!\langle U^\top U - V^\top V, \ U^\top U - V^\top V - U^\top U_{\pi^*} + V^\top V_{\pi^*} \rangle\!\rangle$$

$$= \frac{1}{4}\|U^\top U - V^\top V\|_{\mathrm{F}}^2 + \frac{1}{4}\langle\!\langle U^\top U - V^\top V, \ U^\top U - V^\top V - 2U^\top U_{\pi^*} + 2V^\top V_{\pi^*} \rangle\!\rangle$$

$$= \frac{1}{4}\|U^\top U - V^\top V\|_{\mathrm{F}}^2 + \frac{1}{4}\langle\!\langle U^\top U - V^\top V, \ U^\top U - V^\top V - 2\Delta_U^\top U_{\pi^*} + 2\Delta_V^\top V_{\pi^*} \rangle\!\rangle, \quad (45)$$

where the last step follows from $\Delta_U^\top U_{\pi^*} - \Delta_V^\top V_{\pi^*} = U^\top U_{\pi^*} - V^\top V_{\pi^*}$ since $U_{\pi^*}^\top U_{\pi^*} = V_{\pi^*}^\top V_{\pi^*}$. Note that

$$U^\top U - V^\top V = (U_{\pi^*} + \Delta_U)^\top (U_{\pi^*} + \Delta_U) - (V_{\pi^*} + \Delta_V)^\top (V_{\pi^*} + \Delta_V)$$

$$= U_{\pi^*}^\top \Delta_U + \Delta_U^\top U_{\pi^*} + \Delta_U^\top \Delta_U - V_{\pi^*}^\top \Delta_V - \Delta_V^\top V_{\pi^*} - \Delta_V^\top \Delta_V,$$

where we use $U_{\pi^*}^\top U_{\pi^*} = V_{\pi^*}^\top V_{\pi^*}$ again in the last step. Furthermore, since $U^\top U - V^\top V$ is symmetric, we have

$$\langle\!\langle U^\top U - V^\top V, \ U_{\pi^*}^\top \Delta_U + \Delta_U^\top U_{\pi^*} - V_{\pi^*}^\top \Delta_V - \Delta_V^\top V_{\pi^*} \rangle\!\rangle$$

$$= \langle\!\langle U^\top U - V^\top V, \ 2\Delta_U^\top U_{\pi^*} - 2\Delta_V^\top V_{\pi^*} \rangle\!\rangle.$$

Using these arguments, for the second term in (45), denoted by $T_2$, we have

$$T_2 = \frac{1}{4}\langle\!\langle U^\top U - V^\top V, \ \Delta_U^\top \Delta_U - \Delta_V^\top \Delta_V \rangle\!\rangle.$$

Furthermore, we have

$$4T_2 \leq |\langle\!\langle U^\top U - V^\top V, \ \Delta_U^\top \Delta_U - \Delta_V^\top \Delta_V \rangle\!\rangle| \leq \|U^\top U - V^\top V\|_{\mathrm{F}} \left(\|\Delta_U\|_{\mathrm{F}}^2 + \|\Delta_V\|_{\mathrm{F}}^2\right)$$

$$\leq \left(\|U^\top U - U_{\pi^*}^\top U_{\pi^*}\|_{\mathrm{F}} + \|V^\top V - V_{\pi^*}^\top V_{\pi^*}\|_{\mathrm{F}}\right)\delta$$

$$\leq 2\left(\|U_{\pi^*}\|_{\mathrm{op}}\|\Delta_U\|_{\mathrm{F}} + \|V_{\pi^*}\|_{\mathrm{op}}\|\Delta_V\|_{\mathrm{F}}\right)\delta \leq 2\sqrt{2\sigma_1^* \delta^3}. \quad (46)$$

It remains to find a lower bound of $\|U^\top U - V^\top V\|_{\mathrm{F}}$. The following inequality, which we turn to prove later, is true:

$$\|U^\top U - V^\top V\|_{\mathrm{F}}^2 \geq \|UU^\top - U_{\pi^*}U_{\pi^*}^\top\|_{\mathrm{F}}^2 + \|VV^\top - V_{\pi^*}V_{\pi^*}^\top\|_{\mathrm{F}}^2 - 2\|UV^\top - U_{\pi^*}V_{\pi^*}^\top\|_{\mathrm{F}}^2. \quad (47)$$

Proceeding with the first term in (45) by using (47), we get

$$\frac{1}{4}\|U^\top U - V^\top V\|_F^2 = \frac{1}{8}\|U^\top U - V^\top V\|_F^2 + \frac{1}{8}\|U^\top U - V^\top V\|_F^2$$

$$\geq \frac{1}{8}\|U^\top U - V^\top V\|_F^2 + \frac{1}{8}\|UU^\top - U_{\pi^*}U_{\pi^*}^\top\|_F^2 + \frac{1}{8}\|VV^\top - V_{\pi^*}V_{\pi^*}^\top\|_F^2 - \frac{1}{4}\|UV^\top - U_{\pi^*}V_{\pi^*}^\top\|_F^2$$

$$= \frac{1}{8}\|U^\top U - V^\top V\|_F^2 + \frac{1}{8}\|FF^\top - F_{\pi^*}F_{\pi^*}^\top\|_F^2 - \frac{1}{2}\|UV^\top - U_{\pi^*}V_{\pi^*}^\top\|_F^2, \tag{48}$$

where we let

$$F := \begin{bmatrix} U \\ V \end{bmatrix}, \quad F_{\pi^*} := \begin{bmatrix} U_{\pi^*} \\ V_{\pi^*} \end{bmatrix}.$$

Introduce $\Delta_F := F - F_{\pi^*}$. Recall that $\delta := \|\Delta_U\|_F^2 + \|\Delta_V\|_F^2$. Equivalently $\delta = \|\Delta_F\|_F^2$. We have

$$\|FF^\top - F_{\pi^*}F_{\pi^*}^\top\|_F = \|\Delta_F F_{\pi*}^\top + F_{\pi^*}\Delta_F^\top + \Delta_F \Delta_F^\top\|_F$$

$$\geq \|\Delta_F F_{\pi*}^\top + F_{\pi^*}\Delta_F^\top\|_F - \|\Delta_F\|_F^2 = \|\Delta_F F_{\pi*}^\top + F_{\pi^*}\Delta_F^\top\|_F - \delta.$$

For the first term, we have

$$\|\Delta_F F_{\pi*}^\top + F_{\pi^*}\Delta_F^\top\|_F^2 = 2\|\Delta_F F_{\pi*}^\top\|_F^2 + \langle\!\langle \Delta_F F_{\pi*}^\top, \ F_{\pi^*}\Delta_F^\top \rangle\!\rangle$$

$$\geq 2\sigma_r(F_{\pi^*})^2\|\Delta_F\|_F^2 + \langle\!\langle \Delta_F F_{\pi*}^\top, \ F_{\pi^*}\Delta_F^\top \rangle\!\rangle = 4\sigma_r^*\|\Delta_F\|_F^2 + \langle\!\langle \Delta_F F_{\pi*}^\top, \ F_{\pi^*}\Delta_F^\top \rangle\!\rangle.$$

For the cross term, by the following result, proved in [12] (we also provide a proof in Section B.5 for the sake of completeness), we have $\langle\!\langle \Delta_F F_{\pi*}^\top, \ F_{\pi^*}\Delta_F^\top \rangle\!\rangle \geq 0$.

**Lemma 8.** *When* $\|F - F_{\pi^*}\|_{op} < \sqrt{2\sigma_r^*}$, *we have that* $\Delta_F^\top F_{\pi^*}$ *is symmetric.*

Accordingly, we have $\|FF^\top - F_{\pi^*}F_{\pi^*}^\top\|_F \geq 2\sqrt{\sigma_r^*\delta} - \delta \geq \sqrt{\sigma_r^*\delta}$ under condition $\delta \leq \sigma_r^*$. Plugging this lower bound into (48), we obtain

$$\frac{1}{4}\|U^\top U - V^\top V\|_F^2 \geq \frac{1}{8}\|U^\top U - V^\top V\|_F^2 + \frac{1}{8}\sigma_r^*\delta - \frac{1}{2}\|UV^\top - U_{\pi^*}V_{\pi^*}^\top\|_F^2.$$

Putting (45), (46) and the above inequality together completes the proof.

*Proof of inequality* (47). For the term on the left hand side of (47), it is easy to check that

$$\|U^\top U - V^\top V\|_F^2 = \|UU^\top\|_F^2 + \|VV^\top\|_F^2 - 2\|UV^\top\|_F^2. \tag{49}$$

The property $U_{\pi^*}^\top U_{\pi^*} = V_{\pi^*}^\top V_{\pi^*}$ implies that $\|U_{\pi^*}U_{\pi^*}^\top\|_F = \|V_{\pi^*}V_{\pi^*}^\top\|_F = \|U_{\pi^*}V_{\pi^*}^\top\|_F$. Therefore, expanding those quadratic terms on the right hand side of (47), one can show that it is equal to

$$\|UU^\top\|_F^2 + \|VV^\top\|_F^2 - 2\|U_{\pi^*}^\top U\|_F^2 - 2\|V_{\pi^*}^\top V\|_F^2 + 4\langle\!\langle U_{\pi^*}^\top U, \ V_{\pi^*}^\top V \rangle\!\rangle - 2\|UV^\top\|_F^2. \tag{50}$$

Comparing inequalities (49) and (50), it thus remains to show that

$$-2\|U_{\pi^*}^\top U\|_F^2 - 2\|V_{\pi^*}^\top V\|_F^2 + 4\langle\!\langle U_{\pi^*}^\top U, \ V_{\pi^*}^\top V \rangle\!\rangle \leq 0.$$

Equivalently, we always have $\|U_{\pi^*}^\top U - V_{\pi^*}^\top V\|_F^2 \geq 0$, and thus prove (47).

## A.9 Proof of Lemma 4

First, we turn to prove (23). As

$$\nabla_U \mathcal{G}(U,V) = \frac{1}{2}U(U^\top U - V^\top V), \ \nabla_V \mathcal{G}(U,V) = \frac{1}{2}V(V^\top V - U^\top U),$$

we have

$$\|\nabla_U \mathcal{G}(U,V)\|_F^2 + \|\nabla_V \mathcal{G}(U,V)\|_F^2 \leq \frac{1}{4}\left(\|U\|_{op}^2 + \|V\|_{op}^2\right)\|U^\top U - V^\top V\|_F^2.$$

As $(U,V) \in \mathbb{B}_2(\sqrt{\sigma_1^*})$, we thus have $\|U\|_{op} \leq \|U_{\pi^*}\|_{op} + \|U_{\pi^*} - U\|_{op} \leq 2\sqrt{\sigma_1^*}$, and similarly $\|V\|_{op} \leq 2\sqrt{\sigma_1^*}$. We obtain

$$\|\nabla_U \mathcal{G}(U,V)\|_F^2 + \|\nabla_V \mathcal{G}(U,V)\|_F^2 \leq 2\sigma_1^*\|U^\top U - V^\top V\|_F^2.$$

Now we turn to prove (22). We observe that

$$\nabla_M \mathcal{L}(U, V; S) = M + S - M^* - S^*,$$

where we let $M := UV^\top$. We denote the support of $S, S^*$ by $\Omega$ and $\Omega^*$ respectively. Based on the sparse estimator (4) for computing $S$, $\nabla_M \mathcal{L}(U, V; S)$ is only supported on $\Omega^c$. We thus have

$$\begin{aligned}
\|\nabla_M \mathcal{L}(U, V; S)\|_F &\leq \|\Pi_{\Omega^c \setminus \Omega^*}(M - M^*)\|_F + \|\Pi_{\Omega^c \cap \Omega^*}(M - M^* - S^*)\|_F \\
&\leq \|M - M^*\|_F + \|\Pi_{\Omega^c \cap \Omega^*}(M - M^* - S^*)\|_F.
\end{aligned}$$

It remains to upper bound the second term on the right hand side. Following (37) and (38), we have

$$\|\Pi_{\Omega^c \cap \Omega^*}(M - M^* - S^*)\|_F^2 \leq \sum_{(i,j) \in \Omega^c \cap \Omega^*} \frac{\|u_i\|_2^2}{(\gamma - 1)\alpha d_2} + \frac{\|v_j\|_2^2}{(\gamma - 1)\alpha d_1} \leq \frac{2}{\gamma - 1}\|M - M^*\|_F^2,$$

where the last step is proved in (41). By choosing $\gamma = 2$, we thus conclude that

$$\|\nabla_M \mathcal{L}(U, V; S)\|_F \leq (1 + \sqrt{2})\|M - M^*\|_F.$$

## A.10 Proof of Lemma 6

We denote the support of $\Pi_\Phi(S^*)$, $S$ by $\Omega_o^*$ and $\Omega$. We always have $\Omega_o^* \subseteq \Phi$ and $\Omega \subseteq \Phi$.

In the sequel, we establish several results that characterize the properties of $\Phi$. The first result, proved in Section B.2, shows that the Frobenius norm of any incoherent matrix whose row (or column) space are equal to $L^*$ (or $R^*$) is well preserved under partial observations supported on $\Phi$.

**Lemma 9.** *Suppose $M^* \in \mathbb{R}^{d_1 \times d_2}$ is a rank $r$ and $\mu$-incoherent matrix that has SVD $M^* = L^* \Sigma^* R^{*\top}$. Then there exists an absolute constant $c$ such that for any $\epsilon \in (0, 1)$, if $p \geq c\frac{\mu r \log d}{\epsilon^2 (d_1 \wedge d_2)}$, then with probability at least $1 - 2d^{-3}$, we have that for all $A \in \mathbb{R}^{d_2 \times r}, B \in \mathbb{R}^{d_1 \times r}$,*

$$(1 - \epsilon)\|L^* A^\top + BR^{*\top}\|_F^2 \leq p^{-1}\|\Pi_\Phi(L^* A^\top + BR^{*\top})\|_F^2 \leq (1 + \epsilon)\|L^* A^\top + BR^{*\top}\|_F^2.$$

We need the next result, proved in Section B.3, to control the number of nonzero entries per row and column in $\Omega_o^*$ and $\Phi$.

**Lemma 10.** *If $p \geq \frac{56}{3}\frac{\log d}{\alpha(d_1 \wedge d_2)}$, then with probability at least $1 - 6d^{-1}$, we have*

$$\left||\Phi_{(i,\cdot)}| - pd_2\right| \leq \frac{1}{2}pd_2, \quad \left||\Phi_{(\cdot,j)}| - pd_1\right| \leq \frac{1}{2}pd_1, \quad |\Omega_{o(i,\cdot)}^*| \leq \frac{3}{2}\alpha pd_2, \quad |\Omega_{o(\cdot,j)}^*| \leq \frac{3}{2}\alpha pd_1,$$

*for all $i \in [d_1]$ and $j \in [d_2]$.*

The next lemma, proved in Section B.4, can be used to control the projection of small matrices to $\Phi$.

**Lemma 11.** *There exists constant $c$ such that for any $\epsilon \in (0, 1)$, if $p \geq c\frac{\mu^2 r^2 \log d}{\epsilon^2 (d_1 \wedge d_2)}$, then with probability at least $1 - \mathcal{O}(d^{-1})$, for all matrices $Z \in \mathbb{R}^{d_1 \times d_2}, U \in \mathbb{R}^{d_1 \times r}$ and $V \in \mathbb{R}^{d_2 \times r}$ that satisfy $\|U\|_{2,\infty} \leq \sqrt{\mu r/d_1}, \|V\|_{2,\infty} \leq \sqrt{\mu r/d_2}$, we have*

$$p^{-1}\|\Pi_\Phi(UV^\top)\|_F^2 \leq \|U\|_F^2\|V\|_F^2 + \epsilon\|U\|_F\|V\|_F; \tag{51}$$

$$p^{-1}\|\Pi_\Phi(Z)V\|_F^2 \leq 2\mu r\|\Pi_\Phi(Z)\|_F^2; \tag{52}$$

$$p^{-1}\|U^\top \Pi_\Phi(Z)\|_F^2 \leq 2\mu r\|\Pi_\Phi(Z)\|_F^2. \tag{53}$$

In the remainder of this section, we condition on the events in Lemmas 9, 10 and 11. Now we are ready to prove Lemma 6.

*Proof of Lemma 6.* Using shorthand $M := UV^\top$, we have

$$\nabla_M \widetilde{\mathcal{L}}(U, V; S) = p^{-1}\Pi_\Phi(M + S - M^* - S^*).$$

Plugging it back into the left hand side of (21), we obtain

$$
\langle\!\langle \nabla_M \widetilde{\mathcal{L}}(U, V; S),\ UV^\top - U_{\pi^*} V_{\pi^*}^\top + \Delta_U \Delta_V^\top \rangle\!\rangle
$$
$$
= \frac{1}{p} \langle\!\langle \Pi_\Phi (M + S - M^* - S^*),\ M - M^* + \Delta_U \Delta_V^\top \rangle\!\rangle
$$
$$
\geq \underbrace{\frac{1}{p} \|\Pi_\Phi (M - M^*)\|_{\mathrm{F}}^2}_{T_1} - \underbrace{\frac{1}{p} |\langle\!\langle \Pi_\Phi (S - S^*),\ M - M^* \rangle\!\rangle|}_{T_2} - \underbrace{\frac{1}{p} |\langle\!\langle \Pi_\Phi (M + S - M^* - S^*),\ \Delta_U \Delta_V^\top \rangle\!\rangle|}_{T_3}.
$$
$$
\tag{54}
$$

Next we derive lower bounds of $T_1$, upper bounds of $T_2$ and $T_3$ respectively.

**Lower bound of $T_1$.** We observe that $M - M^* = U_{\pi^*}^* \Delta_V^\top + \Delta_U V_{\pi^*}^\top + \Delta_U \Delta_V^\top$. By triangle inequality, we have

$$
\|\Pi_\Phi(M - M^*)\|_{\mathrm{F}} \geq \|\Pi_\Phi(U_{\pi^*}^* \Delta_V^\top + \Delta_U V_{\pi^*}^\top)\|_{\mathrm{F}} - \|\Pi_\Phi(\Delta_U \Delta_V^\top)\|_{\mathrm{F}}.
$$

Note that when $c \geq a - b$ for $a, b \geq 0$, we always have $c^2 \geq \frac{1}{2} a^2 - b^2$. We thus have

$$
T_1 \geq \frac{1}{2p} \|\Pi_\Phi(U_{\pi^*}^* \Delta_V^\top + \Delta_U V_{\pi^*}^\top)\|_{\mathrm{F}}^2 - \frac{1}{p} \|\Pi_\Phi(\Delta_U \Delta_V^\top)\|_{\mathrm{F}}^2
$$
$$
\geq \frac{1}{2}(1 - \epsilon) \|U_{\pi^*}^* \Delta_V^\top + \Delta_U V_{\pi^*}^\top\|_{\mathrm{F}}^2 - \frac{1}{p} \|\Pi_\Phi(\Delta_U \Delta_V^\top)\|_{\mathrm{F}}^2
$$
$$
\geq \frac{1}{2}(1 - \epsilon) \|M - M^* - \Delta_U \Delta_V^\top\|_{\mathrm{F}}^2 - \|\Delta_U\|_{\mathrm{F}}^2 \|\Delta_V\|_{\mathrm{F}}^2 - 9\epsilon\sigma_1^* \|\Delta_U\|_{\mathrm{F}} \|\Delta_V\|_{\mathrm{F}}
$$
$$
\geq \frac{1}{4}(1 - \epsilon) \|M - M^*\|_{\mathrm{F}}^2 - \frac{1}{2}(1 - \epsilon) \|\Delta_U \Delta_V\|_{\mathrm{F}}^2 - \|\Delta_U\|_{\mathrm{F}}^2 \|\Delta_V\|_{\mathrm{F}}^2 - 9\epsilon\sigma_1^* \|\Delta_U\|_{\mathrm{F}} \|\Delta_V\|_{\mathrm{F}}
$$
$$
\geq \frac{1}{4}(1 - \epsilon) \|M - M^*\|_{\mathrm{F}}^2 - 2\delta^2 - 5\epsilon\sigma_1^*\delta.
$$

where the second step is implied by Lemma 9, the third step follows from (51) in Lemma 11 by noticing that $\|\Delta_U\|_{2,\infty} \leq 3\sqrt{\mu r \sigma_1^* / d_1}$ and $\|\Delta_V\|_{2,\infty} \leq 3\sqrt{\mu r \sigma_1^* / d_1}$, which is further implied by (31).

**Upper bound of $T_2$.** Since $S - S^*$ is supported on $\Omega_0^* \cup \Omega$, we have

$$
pT_2 \leq |\langle\!\langle \Pi_{\Omega_o^* \backslash \Omega}(S^*),\ \Pi_{\Omega_o^* \backslash \Omega}(M - M^*) \rangle\!\rangle| + |\langle\!\langle \Pi_\Omega(S - S^*),\ \Pi_\Omega(M - M^*) \rangle\!\rangle|. \tag{55}
$$

For any $(i, j) \in \Omega$, we have $(S - S^*)_{(i,j)} = (M^* - M)_{(i,j)}$. Therefore, for the second term on the right hand side, we have

$$
|\langle\!\langle \Pi_\Omega(S - S^*),\ \Pi_\Omega(M - M^*) \rangle\!\rangle| \leq \|\Pi_\Omega(M - M^*)\|_{\mathrm{F}}^2 \leq 18\gamma p\alpha\mu r\sigma_1^*\delta, \tag{56}
$$

where the last inequality follows from Lemma 14 and the fact that $|\Omega_{(i,\cdot)}| \leq \gamma p\alpha d_2$, $|\Omega_{(\cdot,j)}| \leq \gamma p\alpha d_1$ for all $i \in [d_1]$, $j \in [d_2]$.

We denote the $i$-th row of $\Pi_\Phi(M - M^*)$ by $u_i$, and we denote the $j$-th column of $\Pi_\Phi(M - M^*)$ by $v_j$. We let $u_i^{(k)}$ denote the element of $u_i$ that has the $k$-th largest magnitude. We let $v_j^{(k)}$ denote the element of $v_j$ that has the $k$-th largest magnitude.

For the first term on the right hand side of (55), we first observe that for $(i, j) \in \Omega_o^* \backslash \Omega$, $|(M^* + S^* - M)_{(i,j)}|$ is either less than the $\gamma p\alpha d_2$-th largest element in the $i$-th row of $\Pi_\Phi(M^* + S^* - M)$, or less than $\gamma p\alpha d_1$-th largest element in the $j$-th row of $\Pi_\Phi(M^* + S^* - M)$. Based on Lemma 10, $\Pi_\Phi(S^*)$ has at most $3p\alpha d_2/2$ nonzero entries per row and at most $3p\alpha d_1/2$ nonzero entries per column. Therefore, we have

$$
|(M^* + S^* - M)_{(i,j)}| \leq \max\left\{ |u_i^{((\gamma-1.5)p\alpha d_2)}|,\ |v_j^{((\gamma-1.5)p\alpha d_1)}| \right\}. \tag{57}
$$

In addition, we observe that

$$|\langle\!\langle \Pi_{\Omega_o^* \setminus \Omega}(S^*),\ \Pi_{\Omega_o^* \setminus \Omega}(M - M^*)\rangle\!\rangle|$$

$$\leq \sum_{(i,j)\in\Omega_o^*\setminus\Omega} |(M^* + S^* - M)_{(i,j)}||(M^* - M)_{(i,j)}| + |(M^* - M)_{(i,j)}|^2$$

$$\leq \left(1 + \frac{\beta}{2}\right)\|\Pi_{\Omega_o^*}(M^* - M)\|_{\mathrm{F}}^2 + \frac{1}{2\beta}\sum_{(i,j)\in\Omega_o^*\setminus\Omega}|(M^* + S^* - M)_{(i,j)}|^2,$$

$$\leq (27 + 14\beta)p\alpha\mu r\sigma_1^*\delta + \frac{1}{2\beta}\sum_{(i,j)\in\Omega_o^*\setminus\Omega}|(M^* + S^* - M)_{(i,j)}|^2, \tag{58}$$

where the second step holds for any $\beta > 0$ and the last step follows from Lemma 14 under the size constraints of $\Omega_o^*$ shown in Lemma 10. For the second term in (58), using (57), we have

$$\sum_{(i,j)\in\Omega_o^*\setminus\Omega}|(M^* + S^* - M)_{(i,j)}|^2 \leq \sum_{(i,j)\in\Omega_o^*}|u_i^{((\gamma-1.5)p\alpha d_2)}|^2 + |v_j^{((\gamma-1.5)p\alpha d_1)}|^2$$

$$= \sum_{i\in[d_1]}\sum_{j\in\Omega_{o(i,\cdot)}^*}|u_i^{((\gamma-1.5)p\alpha d_2)}|^2 + \sum_{j\in[d_2]}\sum_{i\in\Omega_{o(\cdot,j)}^*}|v_j^{((\gamma-1.5)p\alpha d_1)}|^2$$

$$\leq \sum_{i\in[d_1]}\frac{1.5}{\gamma-1.5}\|u_i\|_2^2 + \sum_{j\in[d_2]}\frac{1.5}{\gamma-1.5}\|v_j\|_2^2 \leq \frac{3}{\gamma-1.5}\|\Pi_\Phi(M - M^*)\|_{\mathrm{F}}^2. \tag{59}$$

Moreover, we have

$$\|\Pi_\Phi(M - M^*)\|_{\mathrm{F}}^2 \leq 2\|\Pi_\Phi(U_{\pi^*}\Delta_V^\top + \Delta_U V_{\pi^*}^\top)\|_{\mathrm{F}}^2 + 2\|\Pi_\Phi(\Delta_U\Delta_V^\top)\|_{\mathrm{F}}^2$$

$$\leq 2(1+\epsilon)p\|U_{\pi^*}\Delta_V^\top + \Delta_U V_{\pi^*}^\top\|_{\mathrm{F}}^2 + 2p\|\Delta_U\|_{\mathrm{F}}^2\|\Delta_V\|_{\mathrm{F}}^2 + 18p\epsilon\sigma_1^*\|\Delta_U\|_{\mathrm{F}}\|\Delta_V\|_{\mathrm{F}}$$

$$\leq 4(1+\epsilon)p\left(\|U_{\pi^*}\|_{\mathrm{op}}^2\|\Delta_V\|_{\mathrm{F}}^2 + \|V_{\pi^*}\|_{\mathrm{op}}^2\|\Delta_U\|_{\mathrm{F}}^2\right) + 2p\|\Delta_U\|_{\mathrm{F}}^2\|\Delta_V\|_{\mathrm{F}}^2 + 18p\epsilon\sigma_1^*\|\Delta_U\|_{\mathrm{F}}\|\Delta_V\|_{\mathrm{F}}$$

$$\leq (4 + 13\epsilon)p\sigma_1^*\delta + 2p\delta^2, \tag{60}$$

where the second step follows from Lemma 9 and inequality (51) in Lemma 11. Putting (55)-(60) together, we obtain

$$T_2 \leq (18\gamma + 14\beta + 27)\alpha\mu r\sigma_1^*\delta + \frac{3[(2 + 7\epsilon)\sigma_1^*\delta + \delta^2]}{\beta(\gamma - 1.5)}.$$

**Upper bound of $T_3$.** By Cauchy-Schwarz inequality, we have

$$pT_3 \leq \|\Pi_\Phi(M - M^* + S - S^*)\|_{\mathrm{F}}\|\Pi_\Phi(\Delta_U\Delta_V^\top)\|_{\mathrm{F}}$$

$$\leq \|\Pi_\Phi(M - M^* + S - S^*)\|_{\mathrm{F}}\sqrt{p\|\Delta_U\|_{\mathrm{F}}^2\|\Delta_V\|_{\mathrm{F}}^2 + 9p\epsilon\sigma_1^*\|\Delta_U\|_{\mathrm{F}}\|\Delta_V\|_{\mathrm{F}}}$$

$$\leq \|\Pi_\Phi(M - M^* + S - S^*)\|_{\mathrm{F}}\sqrt{p\delta^2 + 5p\epsilon\sigma_1^*\delta}.$$

where we use (51) in Lemma 11 in the second step.

We observe that $\Pi_\Phi(M - M^* + S - S^*)$ is supported on $\Phi \setminus \Omega$. Therefore, we have

$$\|\Pi_\Phi(M - M^* + S - S^*)\|_{\mathrm{F}} \leq \|\Pi_{\Phi\cap\Omega^c\cap\Phi^{*c}}(M - M^*)\|_{\mathrm{F}} + \|\Pi_{\Phi\cap\Omega^c\cap\Phi^*}(M - M^* - S^*)\|_{\mathrm{F}}$$

$$\leq \|\Pi_\Phi(M - M^*)\|_{\mathrm{F}} + \|\Pi_{\Omega^c\cap\Phi^*}(M - M^* - S^*)\|_{\mathrm{F}}$$

$$\leq \|\Pi_\Phi(M - M^*)\|_{\mathrm{F}} + \sqrt{\frac{3}{\gamma - 1.5}}\|\Pi_\Phi(M - M^*)\|_{\mathrm{F}}$$

$$\leq \left(1 + \sqrt{\frac{3}{\gamma - 1.5}}\right)\sqrt{(4 + 13\epsilon)p\sigma_1^*\delta + 2p\delta^2},$$

where the third step follows from (59), and the last step is from (60). Under assumptions $\gamma = 3$, $\epsilon \leq 1/4$ and $\delta \leq \sigma_1^*$, we have

$$T_3 \leq 3\sqrt{9\sigma_1^*\delta + 2\delta^2}\sqrt{\delta^2 + 5\epsilon\sigma_1^*\delta} \leq 10\sqrt{\sigma_1^*\delta^3} + 23\sqrt{\epsilon\sigma_1^*\delta}.$$

**Combining pieces.** Under the aforementioned assumptions, putting all pieces together leads to

$$\langle\!\langle \nabla_M \widetilde{\mathcal{L}}(U, V; S),\ UV^\top - U_{\pi^*} V_{\pi^*}^\top + \Delta_U \Delta_V^\top \rangle\!\rangle$$
$$\geq \frac{3}{16}\|M - M^*\|_F^2 - (14\beta + 81)\alpha\mu r\sigma_1^*\delta - \left(26\sqrt{\epsilon} + \frac{18}{\beta}\right)\sigma_1^*\delta - 10\sqrt{\sigma_1^*\delta^3} - 2\delta^2.$$

$\square$

### A.11 Proof of Lemma 7

Let $M := UV^\top$. We find that

$$\nabla_U \widetilde{\mathcal{L}}(U, V; S) = p^{-1}\Pi_\Phi (M + S - M^* - S^*)\, V,$$
$$\nabla_V \widetilde{\mathcal{L}}(U, V; S) = p^{-1}\Pi_\Phi (M + S - M^* - S^*)^\top U.$$

Conditioning on the event in Lemma 11, since $(U, V) \in \bar{\mathcal{U}} \times \bar{\mathcal{V}}$, inequalities (52) and (53) imply that

$$\|\nabla_U \widetilde{\mathcal{L}}(U, V; S)\|_F^2 + \|\nabla_V \widetilde{\mathcal{L}}(U, V; S)\|_F^2 \leq \frac{12}{p}\mu r\sigma_1^* \|\Pi_\Phi (M + S - M^* - S^*)\|_F^2.$$

It remains to bound the term $\|\Pi_\Phi (M + S - M^* - S^*)\|_F^2$. Let $\Omega_o^*$ and $\Omega$ be the support of $\Pi_\Phi(S^*)$ and $S$ respectively. We observe that

$$\|\Pi_\Phi (M + S - M^* - S^*)\|_F^2 = \|\Pi_{\Omega_o^* \backslash \Omega} (M - M^* - S^*)\|_F^2 + \|\Pi_{\Phi^* c \cap \Omega^c \cap \Phi} (M - M^*)\|_F^2$$
$$\leq \|\Pi_{\Omega_o^* \backslash \Omega} (M - M^* - S^*)\|_F^2 + \|\Pi_\Phi (M - M^*)\|_F^2.$$

In the proof of Lemma 6, it is shown in (59) that

$$\|\Pi_{\Omega_o^* \backslash \Omega} (M - M^* - S^*)\|_F^2 \leq \frac{3}{\gamma - 1.5}\|\Pi_\Phi(M - M^*)\|_F^2.$$

Moreover, following (60), we have that

$$\|\Pi_\Phi(M - M^*)\|_F^2 \leq 2(1 + \epsilon)p\|U_{\pi^*}\Delta_V^\top + \Delta_U V_{\pi^*}^\top\|_F^2 + 2p\|\Delta_U\|_F^2\|\Delta_V\|_F^2 + 18p\epsilon\sigma_1^*\|\Delta_U\|_F\|\Delta_V\|_F$$
$$\leq 4(1 + \epsilon)p\|M - M^*\|_F^2 + (6 + 4\epsilon)p\|\Delta_U\|_F^2\|\Delta_V\|_F^2 + 18p\epsilon\sigma_1^*\|\Delta_U\|_F\|\Delta_V\|_F$$
$$\leq 4(1 + \epsilon)p\|M - M^*\|_F^2 + (6 + 4\epsilon)p\delta^2 + 9p\epsilon\sigma_1^*\delta.$$

We thus finish proving our conclusion by combining all pieces and noticing that $\gamma = 3$ and $\epsilon \leq 1/4$.

## B  Proofs for Technical Lemmas

In this section, we prove several technical lemmas that are used in the proofs of our main theorems.

### B.1  Proof of Lemma 1

We observe that
$$\|A\|_{\mathrm{op}} = \sup_{x \in \mathbb{S}^{d_1 - 1}} \sup_{y \in \mathbb{S}^{d_2 - 1}} x^\top A y.$$

We denote the support of $A$ by $\Omega$. For any $x \in \mathbb{R}^{d_1}$, $y \in \mathbb{R}^{d_2}$ and $\beta > 0$, we have

$$x^\top A y = \sum_{(i,j)\in\Omega} x_i A_{(i,j)} y_j \leq \sum_{(i,j)\in\Omega} \frac{1}{2}\|A\|_\infty (\beta^{-1}x_i^2 + \beta y_j^2)$$

$$= \frac{1}{2}\|A\|_\infty \left( \sum_i \sum_{j\in\Omega_{(i,\cdot)}} \beta^{-1}x_i^2 + \sum_j \sum_{i\in\Omega_{(\cdot,j)}} \beta y_j^2 \right)$$

$$\leq \frac{1}{2}\|A\|_\infty \left( \alpha d_2 \beta^{-1}\|x\|_2^2 + \alpha d_1 \beta \|y\|_2^2 \right).$$

It is thus implied that $\|A\|_{\mathrm{op}} \leq \frac{1}{2}\alpha(\beta^{-1}d_2 + \beta d_1)\|A\|_\infty$. Choosing $\beta = \sqrt{d_2/d_1}$ completes the proof.

## B.2 Proof of Lemma 9

We define a subspace $\mathcal{K} \subseteq \mathbb{R}^{d_1 \times d_2}$ as

$$\mathcal{K} := \left\{ X \mid X = L^* A^\top + B R^{*\top} \text{ for some } A \in \mathbb{R}^{d_2 \times r}, B \in \mathbb{R}^{d_1 \times r} \right\}.$$

Let $\Pi_\mathcal{K}$ be Euclidean projection onto $\mathcal{K}$. Then according to Theorem 4.1 in [7], under our assumptions, for all matrices $X \in \mathbb{R}^{d_1 \times d_2}$, inequality

$$p^{-1} \| \left( \Pi_\mathcal{K} \Pi_\Phi \Pi_\mathcal{K} - p\Pi_\mathcal{K} \right) X \|_\mathrm{F} \leq \epsilon \|X\|_\mathrm{F} \tag{61}$$

holds with probability at least $1 - 2d^{-3}$.

In our setting, by restricting $X = L^* A^\top + B R^{*\top}$, we have $\Pi_\mathcal{K} X = X$. Therefore, (61) implies that

$$\|\Pi_\mathcal{K} \Pi_\Phi X - pX\|_\mathrm{F} \leq p\epsilon \|X\|_\mathrm{F}.$$

For $\|\Pi_\Phi X\|_\mathrm{F}^2$, we have

$$
\begin{aligned}
\|\Pi_\Phi X\|_\mathrm{F}^2 &= \langle\!\langle \Pi_\Phi X, \ \Pi_\Phi X \rangle\!\rangle = \langle\!\langle \Pi_\Phi X, \ X \rangle\!\rangle \\
&= \langle\!\langle \Pi_\mathcal{K} \Pi_\Phi X, \ X \rangle\!\rangle \leq \|\Pi_\mathcal{K} \Pi_\Phi X\|_\mathrm{F} \|X\|_\mathrm{F} \leq p(1 + \epsilon)\|X\|_\mathrm{F}^2.
\end{aligned}
$$

On the other hand, we have

$$
\begin{aligned}
\|\Pi_\Phi X\|_\mathrm{F}^2 &= \langle\!\langle \Pi_\mathcal{K} \Pi_\Phi X, \ X \rangle\!\rangle = \langle\!\langle \Pi_\mathcal{K} \Pi_\Phi X - pX + pX, \ X \rangle\!\rangle \\
&= p\|X\|_\mathrm{F}^2 - \langle\!\langle X, \ -\Pi_\mathcal{K} \Pi_\Phi X + pX \rangle\!\rangle \\
&\geq p\|X\|_\mathrm{F}^2 - \|X\|_\mathrm{F} \|\Pi_\mathcal{K} \Pi_\Phi X - pX\|_\mathrm{F} \geq p(1 - \epsilon)\|X\|_\mathrm{F}^2.
\end{aligned}
$$

Combining the above two inequalities, we complete the proof.

## B.3 Proof of Lemma 10

We observe that $|\Phi_{(i,\cdot)}|$ is a summation of $d_2$ i.i.d. binary random variables with mean $p$ and variance $p(1 - p)$. By Bernstein's inequality, for any $i \in [d_1]$,

$$\Pr\left[ \left| |\Phi_{(i,\cdot)}| - pd_2 \right| \geq \frac{1}{2}pd_2 \right] \leq 2\exp\left( -\frac{-\frac{1}{2}(pd_2/2)^2}{d_2 p(1-p) + \frac{1}{3}(pd_2/2)} \right) \leq 2\exp\left( -\frac{3}{28}pd_2 \right).$$

By probabilistic union bound, we have

$$\Pr\left[ \sup_{i \in [d_1]} \left| |\Phi_{(i,\cdot)}| - pd_2 \right| \geq \frac{1}{2}pd_2 \right] \leq 2d_1 \exp\left( -\frac{3}{28}pd_2 \right) \leq 2d^{-1},$$

where the last inequality holds by assuming $p \geq \frac{56}{3}\frac{\log d}{d_2}$.

The term $|\Omega^*_{o(i,\cdot)}|$ is a summation of at most $\alpha d_2$ i.i.d. binary random variables with mean $p$ and variance $p(1 - p)$. Again, applying Bernstein's inequality leads to

$$\Pr\left[ |\Omega^*_{o(i,\cdot)}| - \mathbb{E}\left[ |\Omega^*_{o(i,\cdot)}| \right] \geq \frac{1}{2}p\alpha d_2 \right] \leq \exp\left( -\frac{3}{28}p\alpha d_2 \right).$$

Accordingly, by the assumption $p \geq \frac{56}{3}\frac{\log d}{\alpha d_2}$, we obtain

$$\Pr\left[ \sup_{i \in [d_1]} |\Omega^*_{o(i,\cdot)}| - pk \geq \frac{1}{2}pk \right] \leq d_1 \exp\left( -\frac{3}{28}p\alpha d_2 \right) \leq d^{-1}.$$

The proofs for $|\Phi_{(\cdot,j)}|$ and $|\Omega^*_{o(\cdot,j)}|$ follow the same idea.

## B.4 Proof of Lemma 11

According to Lemma 3.2 in [5], under condition $p \geq c_1 \frac{\mu \log d}{d_1 \wedge d_2}$, for any fixed matrix $A \in \mathbb{R}^{d_1 \times d_2}$, we have

$$\|A - p^{-1} \Pi_\Phi A\|_{\mathrm{op}} \leq c_2 \sqrt{\frac{d \log d}{p}} \|A\|_\infty,$$

holds with probability at least $1 - \mathcal{O}(d^{-3})$. Letting $A$ be all-ones matrix, then we have that for all $u \in \mathbb{R}^{d_1}, v \in \mathbb{R}^{d_2}$,

$$\sum_{(i,j) \in \Phi} u_i v_j \leq p \|u\|_1 \|v\|_1 + c_2 \sqrt{pd \log d} \|u\|_2 \|v\|_2.$$

We find that

$$\begin{aligned}
\|\Pi_\Phi(UV^\top)\|_{\mathrm{F}}^2 &\leq \sum_{(i,j) \in \Phi} \|U_{(i,\cdot)}\|_2^2 \|V_{(j,\cdot)}\|_2^2 \\
&\leq p \|U\|_{\mathrm{F}}^2 \|V\|_{\mathrm{F}}^2 + c_2 \sqrt{pd \log d} \sqrt{\sum_{i \in [d_1]} \|U_{(i,\cdot)}\|_2^4} \sqrt{\sum_{j \in [d_2]} \|V_{(j,\cdot)}\|_2^4} \\
&\leq p \|U\|_{\mathrm{F}}^2 \|V\|_{\mathrm{F}}^2 + c_2 \sqrt{pd \log d} \|U\|_{\mathrm{F}} \|V\|_{\mathrm{F}} \|U\|_{2,\infty} \|V\|_{2,\infty} \\
&\leq p \|U\|_{\mathrm{F}}^2 \|V\|_{\mathrm{F}}^2 + c_2 \sqrt{\frac{p \mu^2 r^2 d \log d}{d_1 d_2}} \|U\|_{\mathrm{F}} \|V\|_{\mathrm{F}}.
\end{aligned}$$

By the assumption $p \gtrsim \frac{\mu^2 r^2 \log d}{\epsilon^2 (d_1 \wedge d_2)}$, we finish proving (51).

According to the proof of Lemma 10, if $p \geq c \frac{\log d}{d_1 \wedge d_2}$, with probability at least $1 - \mathcal{O}(d^{-1})$, we have $|\Phi_{(i,\cdot)}| \leq \frac{3}{2} pd_2$ and $|\Phi_{(\cdot,j)}| \leq \frac{3}{2} pd_1$ for all $i \in [d_1]$ and $j \in [d_2]$. Conditioning on this event, we have

$$\begin{aligned}
\|\Pi_\Phi(Z)V\|_{\mathrm{F}}^2 &= \sum_{i \in [d_1]} \sum_{k \in [r]} \langle (\Pi_\Phi(Z))_{(i,\cdot)}, H_{(\cdot,k)} \rangle^2 \\
&\leq \sum_{i \in [d_1]} \sum_{k \in [r]} \|(\Pi_\Phi(Z))_{(i,\cdot)}\|_2^2 \sum_{j \in \Omega_{(i,\cdot)}} V_{(j,k)}^2 \\
&= \|\Pi_\Phi Z\|_{\mathrm{F}}^2 \sum_{j \in \Omega_{(i,\cdot)}} \|V_{(i,\cdot)}\|_2^2 \\
&\leq \|\Pi_\Phi Z\|_{\mathrm{F}}^2 \frac{3}{2} pd_2 \cdot \|V\|_{2,\infty}^2 \leq 2\mu rp \|\Pi_\Phi Z\|_{\mathrm{F}}^2.
\end{aligned}$$

We thus finish proving (52). Inequality (53) can be proved in the same way.

## B.5 Proof of Lemma 8

Recall that we let $F := [U; V]$ and $F_{\pi^*} := [U^*; V^*]Q$ for some matrix $Q \in \mathbb{Q}_r$, which minimizes the following function

$$\|F - [U^*; V^*]Q\|_{\mathrm{F}}^2. \tag{62}$$

Let $F^* := [U^*; V^*]$. Expanding the above term, we find that $Q$ is the maximizer of $\langle\!\langle F, F^*Q \rangle\!\rangle = \mathrm{Tr}(F^\top F^* Q)$. Suppose $F^\top F^*$ has SVD with form $Q_1 \Lambda Q_2^\top$ for $Q_1, Q_2 \in \mathbb{Q}_r$. When the minimum diagonal term of $\Lambda$ is positive, we conclude that the minimizer of (62) is unique and $Q = Q_2 Q_1^\top$. To prove this argument, we note that

$$\mathrm{Tr}(F^\top F^* Q) = \sum_{i \in [r]} \Lambda_{(i,i)} \langle p_i, q_i \rangle,$$

where $p_i$ is the $i$-th column of $Q_1$ and $q_i$ is the $i$-th column of $Q^\top Q_2$. Hence, $\mathrm{Tr}(F^\top F^* Q) \leq \sum_{i \in [r]} \Lambda_{(i,i)}$ and the equality holds if and only if $p_i = q_i$ for all $i \in [r]$ since every $\Lambda_{(i,i)} > 0$. We have $Q_1 = Q^\top Q_2$ and thus finish proving the argument.

Under our assumption $\|F - F_{\pi^*}\|_{\mathrm{op}} < \sqrt{2\sigma_r^*}$, for any nonzero vector $u \in \mathbb{R}^r$, we have

$$\|F^\top F_{\pi^*} u\|_2 \geq \|F_{\pi^*}^\top F_{\pi^*} u\|_2 - \|(F_{\pi^*} - F)^\top F_{\pi^*} u\|_2 \geq (\sqrt{2\sigma_r^*} - \|F_{\pi^*} - F\|_{\mathrm{op}})\|F_{\pi^*} u\|_F > 0.$$

In the second step, we use the fact that the singular values of $F_{\pi^*}$ are equal to the diagonal terms of $\sqrt{2}\Sigma^{*1/2}$. Hence, $F^\top F_{\pi^*}$ has full rank. Furthermore, it implies that $F^\top F^*$ has full rank and only contains positive singular values.

Proceeding with the proved argument, we have

$$F^\top F_{\pi^*} = Q_1 \Lambda Q_2^\top Q_2 Q_1^\top = Q_1 \Lambda Q_1^\top,$$

which implies that $F^\top F_{\pi^*}$ is symmetric. Accordingly, we have $(F - F_{\pi^*})^\top F_{\pi^*}$ is also symmetric.

## C  Supporting Lemmas

In this section, we provide several technical lemmas used for proving our main results.

**Lemma 12.** *For any* $(U^*, V^*) \in \mathcal{E}(M^*)$, $U \in \mathbb{R}^{d_1 \times r}$ *and* $V \in \mathbb{R}^{d_2 \times r}$, *we have*

$$\|UV^\top - U^*V^{*\top}\|_F \leq \sqrt{\sigma_1^*}(\|\Delta_V\|_F + \|\Delta_U\|_F) + \|\Delta_U\|_F \|\Delta_V\|_F,$$

*where* $\Delta_U := U - U^*$, $\Delta_V := V - V^*$.

*Proof.* We observe that $UV^\top - U^*V^{*\top} = U^*\Delta_V^\top + \Delta_U V^{*\top} + \Delta_U \Delta_V^\top$. Hence,

$$\begin{aligned}
\|UV^\top - U^*V^{*\top}\|_F &\leq \|U^*\Delta_V^\top\|_F + \|\Delta_U V^{*\top}\|_F + \|\Delta_U \Delta_V^\top\|_F \\
&\leq \|U^*\|_{\mathrm{op}}\|\Delta_V\|_F + \|V^*\|_{\mathrm{op}}\|\Delta_U\|_F + \|\Delta_U\|_F\|\Delta_V\|_F.
\end{aligned}$$

$\square$

Furthermore, assuming $(U, V) \in \mathcal{U} \times \mathcal{V}$, where $\mathcal{U}$ and $\mathcal{V}$ satisfy the conditions in (19), we have the next result.

**Lemma 13.** *For any* $(i, j) \in [d_1] \times [d_2]$, *we have*

$$|(UV^\top - U^*V^{*\top})_{(i,j)}| \leq 3\sqrt{\frac{\mu r \sigma_1^*}{d_1}}\|\Delta_{V(j,\cdot)}\|_2 + 3\sqrt{\frac{\mu r \sigma_1^*}{d_2}}\|\Delta_{U(i,\cdot)}\|_2 \qquad (63)$$

*Proof.* We observe that

$$\begin{aligned}
|(UV^\top - M^*)_{(i,j)}| &\leq |\langle U^*_{(i,\cdot)}, \Delta_{V(j,\cdot)}\rangle| + |\langle V^*_{(j,\cdot)}, \Delta_{U(i,\cdot)}\rangle| + |\langle \Delta_{U(i,\cdot)}, \Delta_{V(j,\cdot)}\rangle| \\
&\leq \sqrt{\frac{\mu r \sigma_1^*}{d_1}}\|\Delta_{V(j,\cdot)}\|_2 + \sqrt{\frac{\mu r \sigma_1^*}{d_2}}\|\Delta_{U(i,\cdot)}\|_2 + \frac{1}{2}\|\Delta_U\|_{2,\infty}\|\Delta_{V(j,\cdot)}\|_2 + \frac{1}{2}\|\Delta_V\|_{2,\infty}\|\Delta_{U(i,\cdot)}\|_2.
\end{aligned}$$

By noticing that

$$\|\Delta_U\|_{2,\infty} \leq \|U^*\|_{2,\infty} + \|U\|_{2,\infty} \leq 3\sqrt{\frac{\mu r \sigma_1^*}{d_1}}, \quad \|\Delta_V\|_{2,\infty} \leq \|V^*\|_{2,\infty} + \|V\|_{2,\infty} \leq 3\sqrt{\frac{\mu r \sigma_1^*}{d_2}},$$

we complete the proof. $\square$

Lemma 13 can be used to prove the following result.

**Lemma 14.** *For any* $\alpha \in [0, 1]$, *suppose* $\Omega \subseteq [d_1] \times [d_2]$ *satisfies* $|\Omega_{(i,\cdot)}| \leq \alpha d_2$ *for all* $i \in [d_1]$ *and* $|\Omega_{(\cdot,j)}| \leq \alpha d_1$ *for all* $j \in [d_2]$. *Then we have*

$$\|\Pi_\Omega(UV^\top - U^*V^{*\top})\|_F^2 \leq 18\alpha\mu r \sigma_1^*(\|\Delta_V\|_F^2 + \|\Delta_U\|_F^2).$$

*Proof.* Using Lemma 13 for bounding each entry of $UV^\top - U^*V^{*\top}$, we have that

$$\|\Pi_\Omega(UV^\top - U^*V^{*\top})\|_F^2 \le \sum_{(i,j)\in\Omega} |(UV^\top - U^*V^{*\top})_{(i,j)}|^2$$

$$\le \sum_{(i,j)\in\Omega} \frac{18\mu r\sigma_1^*}{d_1}\|\Delta_{V(j,\cdot)}\|_2^2 + \frac{18\mu r\sigma_1^*}{d_2}\|\Delta_{U(i,\cdot)}\|_2^2$$

$$\le \sum_j \sum_{i\in\Omega_{(\cdot,j)}} \frac{18\mu r\sigma_1^*}{d_1}\|\Delta_{V(j,\cdot)}\|_2^2 + \sum_i \sum_{j\in\Omega_{(i,\cdot)}} \frac{18\mu r\sigma_1^*}{d_2}\|\Delta_{U(i,\cdot)}\|_2^2$$

$$\le 18\alpha\mu r\sigma_1^*(\|\Delta_V\|_F^2 + \|\Delta_U\|_F^2).$$

$\square$

Denote the $i$-th largest singular value of matrix $M$ by $\sigma_i(M)$.

**Lemma 15** (Lemma 5.14 in [24]). *Let $M_1, M_2 \in \mathbb{R}^{d_1\times d_2}$ be two rank $r$ matrices. Suppose they have SVDs $M_1 = L_1\Sigma_1 R_1^\top$ and $M_2 = L_2\Sigma_2 R_2^\top$. Suppose $\|M_1 - M_2\|_{op} \le \frac{1}{2}\sigma_r(M_1)$. Then we have*

$$d^2(L_2\Sigma_2^{1/2}, R_2\Sigma_2^{1/2}; L_1\Sigma_1^{1/2}, R_1\Sigma_1^{1/2}) \le \frac{2}{\sqrt{2}-1} \frac{\|M_2 - M_1\|_F^2}{\sigma_r(M_1)}.$$

## D  Parameter Settings and More Results for FB Separation Experiments

We approximate the FB separation problem by the RPCA framework with $r = 10$, $\alpha = 0.2$, $\mu = 10$. Our algorithmic parameters are set as $\gamma = 1$, $\eta = 1/(2\hat{\sigma}_1^*)$, where $\hat{\sigma}_1^*$ is an estimate of $\sigma_1^*$ obtained from the initial SVD. The parameters of AltProj are kept as provided in the default setting. For IALM, we use the tradeoff paramter $\lambda = 1/\sqrt{d_1}$, where $d_1$ is the number of pixels in each frame (the number of rows in $Y$).

Note that both IALM and AltProj use the stopping criterion

$$\|Y - M_t - S_t\|_F/\|Y\|_F \le 10^{-3}.$$

Our algorithm for the partial observation setting never explicitly forms the $d_1$-by-$d_2$ matrix $M_t = U_tV_t^\top$, which is favored in large scale problems, but also renders the above criterion inapplicable. Instead, we use the following stopping criterion

$$\frac{\|U_{t+1} - U_t\|_F^2 + \|V_{t+1} - V_t\|_F^2}{\|U_t\|_F^2 + \|V_t\|_F^2} \le 4\times 10^{-4}.$$

This rule checks whether the iterates corresponding to low-rank factors become stable. In fact, our stopping criterion seems more natural and practical because in most real applications, matrix $Y$ cannot be strictly decomposed into low-rank $M$ and sparse $S$ that satisfy $Y = M + S$. Instead of forcing $M + S$ to be close to $Y$, our rule relies on seeking a robust subspace that captures the most variance of $Y$.

Figure 3 shows the recovery results for several more frames. Our algorithms enjoy better running time and outperform AltProj and IALM in separating persons from the background images.

Figure 3: More results of FB separation in *Restaurant* and *ShoppingMall* videos. The leftmost images are original frames. The right four columns show the results from our algorithms with $p = 1$, $p = 0.2$, AltProj [21], and IALM [20]. The runtime of each algorithm is written in the title.