[Reviews · NeurIPS 2016]

Reviewer 1

Summary

The paper presents two algorithms for RPCA under full and partial observation. Both algorithms are based on an initialization step that (1) performs an elementwise truncation on the input matrix and (2) performs a singular value truncation on the singular vectors of the input matrix. From this point on a gradient descent scheme is followed on each of the right / left factors. Authors claim their algorithm has a complexity of r = rank of the true matrix better than state-of-the-art (fully observed case). This is a dependence not on the ambiant dimension but on the rank of the unknown matrix, that we assume to be small. But in the case we face a rank of ~ 100 this result is an improvement of ~ 100 times in complexity. The algorithm is based on first order steps on the factors. Each step is a standard operation on the right / left factors and convergence of the gradient descent scheme on the nonconvex objective is guaranteed thanks to the smart initialization step.

Qualitative Assessment

I am wandering if the (r^4 d) rate has any reason to be believed optimal. The same for r d^2. Is the quadratic dependence on d something one can hope to get rid of?

Confidence in this Review

2-Confident (read it all; understood it all reasonably well)


Reviewer 2

Summary

The authors propose an iterative algorithm for robust PCA in both the fully and partially observed setting. The paper's claimed contribution is a runtime better than prior art, in addition to provable robustness guarantees which are not optimal but close to the known optimum. By "robust PCA" the authors mean that the matrix of interest has been additively contaminated by a sparse outlier matrix whose support is deterministic and unknown, obeying the row and column sparsity condition. For the partially observed case, the authors assume i.i.d erasures of the matrix entries. In the paper, the authors review some prior art, define the algorithms and state main results regarding convergence of the algorithms. A small simulation study is reported and a small real-world experiment is conducted.

Qualitative Assessment

overall: the results are relevant and, as far as I can tell, novel. this paper would be a welcome addition to the robust-PCA literature. The paper is well-organized and clear. The technical contribution appears sound. major comments: - the authors gloss other the fact that the algorithm's step size is determined by the unknown quantity sigma^*_1 (the largest signal singular value). I recommend that the authors explain how the step size should be chosen in practice and what happens to the algorithm if a wrong step size is chosen. - The authors gloss over implementation details for the numerical experiments. I recommend that the authors explain in more detail how experiments were conducted (e.g. choice of step size, etc) - for the partial observations case, the error probability is considerable (1-O(1/d)) and does not diminish with iterations. This should be made clear in the into and the abstract. minor comments: - Abstract: define epsilon in the complexity formulae advertised. (r denoting rank, d dimension, epsilon denoting estimation error) - p.2 line 82: "conve" - algorithm 1: define \mathcal{L} or refer to the definition. same for algorithm 2 and \tilde\mathcal{L}}

Confidence in this Review

2-Confident (read it all; understood it all reasonably well)


Reviewer 3

Summary

This paper introduces a new non-convex algorithm for robust PCA in the fully and partially observed settings based on the projected gradient method. In particular, the authors introduce a new sparse estimator for the setting of deterministic corruptions. This estimator keeps the elements of a given matrix that are simultaneously among the largest fraction entries in the corresponding row and column. The projected gradient method based on this sparse estimator is claimed to show linear convergence. For the partial observations case (i.e. large number of missing entries), the proposed algorithm is shown to succeed, and when the rank is small compared to the dimension of the observed matrix, the computational cost is linear. Two concerte algorithms are proposed both for the fully and partially observed settings (Sec. 3), and their analysis is provided in Sec. 4. The detailed proofs are given in the supplementary material. The algorithms are applied on synthetic and on real datasets, and are compared with state of the arts methods including the inexact augmented lagrange multiplier and the alternating projection methods (Sec. 5).

Qualitative Assessment

The paper is well motivated and written, and is of high quality technical material. It addresses an important question in unsupervised machine learning, that is designing fast algorithms for dimensionality reduction in the presence of corrupted and missing observations. The technical results seem to be sound, and the experiments are convincing. Some minor remarks: Abstract: L1. « The » appears two times. L83: conve->convex? Eq (2). I found it hard to understand the otherwise part. Why not simply use the Bernoulli model as introduced in the Candes and Recht paper?

Confidence in this Review

2-Confident (read it all; understood it all reasonably well)


Reviewer 4

Summary

This paper presents an efficient algorithm for the problem of robust PCA. The key idea is estimating a good initial solution followed by gradient descent for improvement. Compared to a recent (non-convex) robust PCA solver, the computational complexity is reduced by a factor of "r", where "r" denotes the rank of the true matrix. The algorithm is straightforward, easy to follow. The main technical contribution is the linear convergence guarantee which appears novel.

Qualitative Assessment

This is an interesting work that presents the perhaps first theoretical guarantee for a widely used optimization technique of the problem of robust PCA. I think it has potential and should be somewhere between poster and oral. During the rebuttal period, I would suggest the authors to address the following minor concerns. - It appears to me that the algorithms (Algorithm 1 and Algorithm 2) is easy to follow. That been said, it is not clear how one is able to know the true rank "r" and the corruption fraction "alpha". The projection step (Step 7 and Step 8) requires a knowledge of the incoherence parameter. How to estimate it? If one only knows some upper bound on the rank and corruption fraction (which should be the practical case), is it possible to show that all the theoretical results still hold? - Theorem 1 states that the initial deviation for "(U_0, V_0)" is upper bounded by a quantity which is proportional to "sqrt(r/kappa)". This means when "r" is large, the initial estimate is not good enough. Given this, why does the algorithm still succeed by projecting the iterates onto the space of "mathcal{U}" and "mathcal{V}" which are determined by "U_0" and "V_0", respectively? Please give more intuition on it. - The sample complexity is not optimal here. The authors claim that it can be improved by utilizing the output of [21]. For the sake of concreteness, the authors should frame such an argument more rigorously, e.g., "if we apply the AltProj[21] for initialization, our Theorem 1 can be strengthened to d(U_0, V_0; U^*, V^*) < ... In this way, we obtain an improved robustness result ..." Otherwise, it is tricky to check the correctness of Remark 2. - When analyzing the computational complexity and robustness, the authors presume that "kappa" is a constant and compare the theoretical result to state-of-the-art solvers. This is somewhat unfair since "kappa" might scale with the dimension. A good practice is carefully discuss when the presented algorithm dominates previous solvers and when it does not, e.g., "when kappa = O(1), we observe that our algorithm is an order of magnitude faster ... when kappa = O(d), we find that ...". It would be better to provide several examples where "kappa" behaves like a constant. --------------------------------- Updates: The author feedback is convincing. It is a nice work to me. A minor comment: It seems that another parallel work [A] also considers online alternating minimization. It would be good if the authors could comment on the difference. [A] Provable Efficient Online Matrix Completion via Non-convex Stochastic Gradient Descent, arXiv:1605.08370

Confidence in this Review

2-Confident (read it all; understood it all reasonably well)


Reviewer 5

Summary

This paper introduces a new algorithm for robust PCA via gradient descent, which is fast and has small memory fingerprint. The authors showed theoretical guarantee of this algorithm, under certain sparsity assumptions. The results are interesting both theoretically and practically.

Qualitative Assessment

This paper presents a novel algorithm for robust PCA that is interesting both in theory and in practice. A few minor comments that may help improve this paper: 1. The projection onto sets U and V are straightforward, however it is still interesting to provide the detail here for the sake of completeness. 2. The theoretical results are very impressive. A drawback is that the algorithm can only allow a sparsity level of O(1/(\mu r^1.5)), instead of the optimal O(1/(\mu r)). This however, is somewhat made up for by having a much faster run-time. Can the authors comment on potential improvement in this aspect? Is this a fundamental limitation of gradient descent, or a limitation of the proof technique? I suspect the latter, since it seems from the experiments section that gradient descent succeeds for alpha as large as 1/r. 3. Is the advantage of gradient descent over alternating projection universal? Can you comment on relevant results in other applications (matrix completion, etc.)?

Confidence in this Review

2-Confident (read it all; understood it all reasonably well)


Reviewer 6

Summary

The paper consider the problem of robust PCA in the fully and partially observation case. The authors propose an algorithms, which alternatively update an estimator for sparse corruption and the low-rank matrix on the factorized space (projected gradient descent) in both fully and partially observation models. The author theoretically show that proposed gradient method succeeds with a better computational complexity by a factor of r, compared to the best previous algorithm (partial observation case). Finally the authors provide some numeric results on synthetic datasets to support their theoretical results. Furthermore the authors applied the proposed algorithm to the task of foreground -background separation in a video. They show some superior experimental results compared to previous known AltProj and IALM on Restaurant and ShoppingMall datasets.

Qualitative Assessment

1. The sets \mathcal{U} and \mathcal{V} defined in Page 5 only relates to the initial estimate U_0 and V_0. Is it better to consider other better sets. For example, these sets are related to the last estimate of U and V. 2. Is the constraint term G really necessary in practice or it is required for the sake of proof. 3. Could you recheck the identity (T2) above equation (46). update ----- The rebuttal is satisfying, though the author leave most of my questions as future work.

Confidence in this Review

2-Confident (read it all; understood it all reasonably well)